# Genuine multipartite entanglement in time

Simon Milz[1,2★] , Cornelia Spee[1,3] , Zhen-Peng Xu[3†] ,
Felix A. Pollock[2] , Kavan Modi[2] and Otfried Gühne[3]

**1** Institute for Quantum Optics and Quantum Information,
Austrian Academy of Sciences, Boltzmanngasse 3, 1090 Vienna, Austria
**2** School of Physics and Astronomy, Monash University, Clayton, Victoria 3800, Australia
**3** Naturwissenschaftlich-Technische Fakultät, Universität Siegen,
Walter-Flex-Straße 3, 57068 Siegen, Germany

★ simon.milz@oeaw.ac.at, † zhen-peng.xu@uni-siegen.de

## Abstract

While spatial quantum correlations have been studied in great detail, much less is known about the genuine quantum correlations that can be exhibited by *temporal* processes. Employing the quantum comb formalism, processes in time can be mapped onto quantum states, with the crucial difference that temporal correlations have to satisfy causal ordering, while their spatial counterpart is not constrained in the same way. Here, we exploit this equivalence and use the tools of multipartite entanglement theory to provide a comprehensive picture of the structure of correlations that (causally ordered) temporal quantum processes can display. First, focusing on the case of a process that is probed at two points in time – which can equivalently be described by a tripartite quantum state – we provide necessary as well as sufficient conditions for the presence of bipartite entanglement in different splittings. Next, we connect these scenarios to the previously studied concepts of quantum memory, entanglement breaking superchannels, and quantum steering, thus providing both a physical interpretation for entanglement in temporal quantum processes, and a determination of the resources required for its creation. Additionally, we construct explicit examples of W-type and GHZ-type genuinely multipartite entangled two-time processes and prove that genuine multipartite entanglement in temporal processes can be an emergent phenomenon. Finally, we show that genuinely entangled processes across multiple times exist for any number of probing times.



# 1 Introduction

Correlations form the basis for scientific inferences about the world. They are used, amongst others, to detect (and discern different types of) causal relations [1–3], to distinguish theories that abide by local realism from those that do not [4–6], and to test if quantum mechanics satisfies the assumptions of non-invasive measurements and realism *per se* [7–9]. The most

striking type of genuine quantum correlations is entanglement [10,11], which is a prerequisite for the violation of Bell inequalities [12] and quantum steering [13–15], two phenomena that lie outside of what is possible by means of classical correlations. Additionally, entanglement provides an advantage in information processing tasks and is a fundamental resource in many quantum information protocols [16].

There have been many attempts to import various facets of spatial quantum correlations to the temporal domain. Most notably, the violation of Leggett-Garg type inequalities [7–9] were introduced to capture quantum correlations in time in analogy to a Bell-type setup. That is, when a single quantum system is probed at different points *in time* the resulting correlations can also go beyond what is possible in classical theories. However, Bell-type setups in time are fraught with difficulties as it is easy to construct fully classical, but invasive, setups that too can violate these inequalities maximally [17,18].

The interpretational issues are less problematic for the case of 'entanglement in time' and thus it is possible to classify and attribute operational meaning to genuinely quantum temporal correlations. In particular, the quantum comb formalism [19, 20] allows one to express *any* temporal quantum process in terms of a multipartite quantum state – called a quantum comb – where each time the process is probed at corresponds to two Hilbert spaces. This is most transparent by means of the so-called Choi-Jamiołkowski isomorphism (CJI), which maps any (multi-time) quantum process to a (many-body) quantum state. Consequently, both spatial and temporal correlations can be analysed in one common framework, enabling the study of temporal correlations on the same mathematical footing as the analysis of spatial ones. In addition, as the respective correlations have a direct interpretation in terms of the properties of the underlying process, it allows one to provide a clear-cut meaning to statements like 'different points in time are entangled with each other'[1].

The quantum combs framework is naturally suited for describing multi-time quantum stochastic processes [2, 23], as well as more exotic processes that lack global causal ordering [24, 25]. In each case quantifying quantum resources has significant operational value. For instance, for the former, the requisite entangling resources naturally relate to the quantum complexity of a stochastic process. Alternatively, having access to naturally occurring or engineered entanglement in time within quantum devices could help to enhance their performance [26]. For the latter case, simulating causally indefinite processes requires spatial entanglement, tying together quantum correlations and exotic temporal phenomena [27,28].

While the isomorphism between quantum states and processes enables the systematic study of temporal correlations, it comes with two caveats; on the one hand, quantum states that correspond to quantum processes have to encapsulate the causal ordering of the process they describe. This means that measurements made at a later time cannot influence the statistics at earlier times, a requirement that imposes a hierarchy of trace conditions on quantum combs [19]. Accordingly, known results on the existence of multipartite quantum states that satisfy desired entanglement properties cannot straightforwardly be applied to quantum combs, as the set of states that describe spatial scenarios does not coincide with the set of states that corresponds to temporal processes [29]. Given these additional constraints, it is then natural to ask, what types of entanglement can exist in combs, and if there are genuinely multipartite entangled combs for any number of times.

On the other hand, the respective interpretations of the observed correlations fundamentally differ in the spatial and the temporal case. While quantum states 'only' describe the correlations between measurements on spatially separated parties, in a quantum comb correlations

---

[1]Within the related framework of consistent histories (CH) [21], correlations as they are displayed in the violation of Leggett-Garg inequalities have consequently been dubbed 'entanglement in time' [22]. As the CH approach as used in [22] only ascribes one Hilbert space to each point in time, the results presented in this paper, as well as their interpretation fundamentally differ from the ones provided in [22].

between different sets of parties have different interpretations. For example, in the simplest case, depending on the involved parties, entanglement can mean that two parties share a quantum state, share a non-entanglement breaking (EB) channel, or possess the ability to transmit quantum memory. Each of these cases has been analysed individually in the literature; phrased in the language of quantum casual modelling, the former two cases amount to a quantum common cause and a quantum direct cause, respectively [2]. In Refs. [30, 31] the authors provided an example of a process that constitutes a superposition of common cause and direct cause scenarios, something, that is unattainable in quantum mechanics. The idea of a quantum memory and its connection to entanglement properties of the underlying comb was introduced in [32].

Here, we provide a systematic study of the entanglement features – both bipartite and multipartite – a quantum comb can display, analyse in detail the respectively necessary and sufficient properties of the underlying dynamics for the presence of different types of entanglement in combs. Put differently, we analyze both what it means in a physical sense for a comb to be entangled, and work out the respective resources that would be required in order to create such different types of entanglement in temporal processes. Specifically, after analysing the bipartite entanglement properties of combs on two times (i.e., defined on three Hilbert spaces), we show that there exist genuinely multipartite entangled combs for any number of times, and provide explicit examples of both of W- and GHZ-type entangled combs on two times. Additionally, we show that in the temporal case – in analogy to its spatial counterpart [33] – there exist processes with entanglement as an emergent quality, i.e., genuinely multipartite entangled states that do not display entanglement in their marginals even when one allows for conditioning. Along the way, we provide explicit circuits for each of the discussed cases, and relate them to existing phenomena discussed in the literature, like entanglement breaking superchannels [34], channel steering [35], as well as the aforementioned concepts of genuine quantum memory and the superposition of direct and common causes. In this way we provide a comprehensive picture of the entanglement properties of temporal processes.

## 2 Preliminaries

### 2.1 Preliminaries: Quantum combs

Throughout this article, we envision the following setup: An experimenter has access to a system – considered to be finite dimensional – which they can manipulate (i.e., transform, measure, discard, etc.) at successive points in time $t_1 < t_2 < \cdots < t_{n+1}$. In between these points, the system evolves freely, potentially interacting with degrees of freedom (henceforth dubbed *environment*) that are out of the control of the experimenter. In the most general case, this free evolution is described by a quantum channel, that is, a completely positive trace preserving (CPTP) map (see Fig. 1). Due to the interaction with the environment, the resulting multi-time statistics, or, equivalently, the quantum stochastic process that the experimenter probes, can display complex memory effects that can go beyond what is possible in classical physics [30–32].

Mathematically, every manipulation the experimenter can perform on the system corresponds to a trace non-increasing completely positive (CP) map. For example, at each time $t_j$, the system of interest could be measured in the computational basis, yielding outcomes $\{x_j\}$. This measurement leaves the original state of the system in the state $|x_j\rangle\langle x_j|$ and is described by the CP map $\mathcal{P}_{x_j}[\rho] = \langle x_j|\rho|x_j\rangle |x_j\rangle\langle x_j|$, where the probability to actually obtain outcome $x_j$ is given by $\mathrm{tr}(\mathcal{P}_{x_j}[\rho])$. More generally, in order to gain different information about the underlying quantum stochastic process, the experimenter could choose to probe it in different

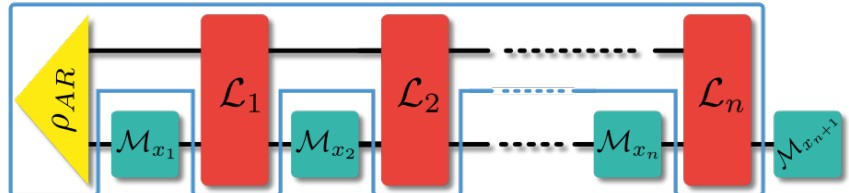

Figure 1: **Temporal quantum process.** A system of interest is probed sequentially at times $t_1, t_2, \ldots$. Initially, the system can be in a state $\rho_{AR}$ that is correlated with degrees of freedom (labelled by $R$) that are outside the experimenter's control. Each measurement corresponds to a map $\mathcal{M}_i$. In between measurements, the system and the environment together undergo a free evolution given by CPTP maps $\mathcal{L}_1, \ldots, \mathcal{L}_n$. The resulting process is fully described by a comb $\mathcal{T}_{n+1}$ (depicted by the blue outline).

ways. For example, instead of measuring in the computational basis, the experimenter could use a *positive operator valued measure* (POVM), and, upon observing outcome $x_j$, feed forward a quantum state $\eta_{x_j}$. In this case, the map corresponding to the experimental manipulation would be given by $\mathcal{M}_{x_j}[\rho] = \mathrm{tr}(E_{x_j}\rho)\eta_{x_j}$, where $E_{x_j}$ is the POVM element corresponding to the outcome $x_j$.

In the most general case, at each time $t_j$ the experimenter could choose a general instrument $\mathcal{I}_j = \{\mathcal{M}_{x_j}\}$, i.e., a collection of CP maps that add up to a CPTP map $\mathcal{M}_j$, such that every outcome $x_j$ they observe corresponds to a transformation of the system given by $\mathcal{M}_{x_j}$. Then, denoting the free system-environment evolution between times $t_j$ and $t_{j-1}$ by $\mathcal{L}_j$, the probability for the experimenter to observe outcomes $x_{n+1}, \ldots, x_1$ having used the instruments $\mathcal{I}_{n+1}, \ldots, \mathcal{I}_1$ at times $t_{n+1}, \ldots, t_1$ is given by

$$
\begin{aligned}
\mathbb{P}(x_{n+1}, \ldots, x_1 | \mathcal{I}_{n+1}, \ldots, \mathcal{I}_1) &= \mathrm{tr}(\mathcal{M}_{x_{n+1}} \circ \mathcal{L}_n \circ \cdots \circ \mathcal{L}_1 \circ \mathcal{M}_{x_1}[\rho_{AR}]) \\
&=: \mathrm{tr}\{\mathcal{T}_{n+1}[\mathcal{M}_{x_{n+1}}, \ldots, \mathcal{M}_{x_1}]\},
\end{aligned}
\tag{1}
$$

where the maps $\{\mathcal{L}_n, \ldots, \mathcal{L}_1\}$ describe the free system-environment evolution in between measurements and we have defined the multilinear functional $\mathcal{T}_{n+1}$ (see Fig. 1). This functional (or slight variations thereof) appears under varying names in various fields of quantum information theory and beyond. Depending on the context, it is called a *quantum comb* [19, 20, 36] (in the study of higher order quantum maps), *process tensor* [23, 37, 38] and *causal automata/non-anticipatory channels* [39, 40] (when concerned with general open quantum processes with memory), *causal box* [41] (when quantum networks with modular elements are investigated), *operator tensor* [42, 43] and *superdensity matrix* [44] (in the field of quantum information in general relativistic space-time), *process matrix* [2, 3, 24, 25] (when used for quantum causal modelling), and *quantum strategy* (in the context of quantum games [45]). Following Refs. [19, 20, 36], we will call $\mathcal{T}_{n+1}$ a quantum comb (or simply comb). Importantly, combs can encapsulate *any* causally ordered quantum process [46], and are thus the mathematical framework for the field of quantum causal modelling [2, 3].

As can be seen from Eq. (1), $\mathcal{T}_{n+1}$ depends on the initial system-environment state $\rho_{AR}$ as well as the intermediate maps $\{\mathcal{L}_j\}$ and contains all statistical information that can be inferred from an underlying process when probing it at times $t_1, \ldots, t_{n+1}$. As such, it contains all spatio-temporal correlations – and thus all causal relations – that the quantum stochastic process of interest can exhibit. Additionally, due to the linearity of quantum mechanics, $\mathcal{T}_{n+1}$ can be reconstructed with a finite number of measurements [23]. These latter two properties of combs are analogous to those of *quantum states*, with the difference that the latter only contain all inferrable *spatial* joint probabilities, while the former contains all *spatio-temporal* correlations for measurements that can be separated both in space and time.

Employing the Choi-Jamiołkowski isomorphism [47–49] we can make this analogy more transparent. Specifically, each of the CP maps $\mathcal{M}_{x_j}$ transforms input (i) states $\rho \in \mathcal{B}(\mathcal{H}_j^{\text{i}})$ to output (o) states $\rho' = \mathcal{M}_{x_j}[\rho] \in \mathcal{B}(\mathcal{H}_j^{\text{o}})$, where $\mathcal{B}(\mathcal{H}_j^{\text{y}})$ denotes the space of bounded linear operators on $\mathcal{H}_j^{\text{y}}$, with $\text{y} \in \{\text{i}, \text{o}\}$. Via the CJI, any such map corresponds to a matrix $M_{x_j} \in \mathcal{B}(\mathcal{H}_j^{\text{o}} \otimes \mathcal{H}_j^{\text{i}})$ (see App. A for details on the CJI), where we adapt the convention that maps are denoted by calligraphic letters, and their Choi matrices by upright ones.

Importantly, the respective output space can also be trivial, i.e., $\mathcal{H}_j^{\text{o}} \cong \mathbb{C}$, which is the case when the system of interest is discarded after the measurement (see below). A map $\mathcal{M}_{x_j}$ is CP iff its Choi matrix is positive, while it is trace preserving (TP) iff its Choi matrix satisfies $\text{tr}_{j^{\text{o}}}(M_{x_j}) = \mathbb{1}_{j^{\text{i}}}$. Here, $\text{tr}_x^{\text{y}}$ ($\mathbb{1}_x^{\text{y}}$) denotes the partial trace over (identity matrix on) the Hilbert space $\mathcal{H}_x^{\text{y}}$.

While we will mostly consider cases where the input and output dimensions of the respective maps coincide (except for the last time $t_{n+1}$, see below), and where the size of the considered system does not vary with time, for better bookkeeping, we always distinguish between the input and output space, and additionally label the respective Hilbert spaces with the time $t_j$ they correspond to. Employing the CJI, Eq. (1) can be rewritten as

$$\mathbb{P}(x_{n+1}, \ldots, x_1 | \mathcal{J}_{n+1}, \ldots, \mathcal{J}_1) = \text{tr}[(M_{x_{n+1}}^{\text{T}} \otimes \cdots \otimes M_{x_1}^{\text{T}}) \Upsilon_{n+1}], \quad (2)$$

where $\bullet^{\text{T}}$ denotes the transposition. $\Upsilon_{n+1} \in \mathcal{B}(\mathcal{H}_{n+1}^{\text{i}} \otimes \mathcal{H}_n^{\text{o}} \otimes \cdots \otimes \mathcal{H}_1^{\text{o}} \otimes \mathcal{H}_1^{\text{i}})$ is the Choi matrix of $\mathcal{T}_{n+1}$, deviating from our normal naming convention for maps and their Choi states to avoid confusion with the transposition and to conform with the notation used in the literature [23, 50]. As the evolution after the final time $t_{n+1}$ is not of interest, without loss of generality, the final instrument $\mathcal{J}_{n+1}$ is a POVM (i.e., it has a trivial output space), implying $\sum_{x_{n+1}} M_{x_{n+1}} = \mathbb{1}_{n+1}^{\text{i}}$. In slight abuse of notation, in what follows, whenever there is no risk of confusion, we will call both $\mathcal{T}_{n+1}$ and its Choi matrix $\Upsilon_{n+1}$ the comb of the quantum process at hand.

It is important to stress the similarity of Eq. (2) to the Born rule. The joint probability distributions for measuring a spatially separated multipartite quantum state $\rho$ with POVM elements $E_{x_1}, \ldots, E_{x_{n+1}}$ corresponding to the outcomes of spatially separated parties $1, \ldots, n+1$ is given by

$$\mathbb{P}(x_{n+1}, \ldots, x_1 | \mathcal{J}_{n+1}, \ldots, \mathcal{J}_1) = \text{tr}[(E_{x_{n+1}} \otimes \cdots \otimes E_{x_1}) \rho]. \quad (3)$$

This is akin to Eq. (2), which thus has been dubbed generalized Born rule for temporal processes [51, 52]. $\Upsilon_{n+1}$ can hence be considered a quantum state *in time*, and it is natural to ask what kinds of correlations it can display, and what kinds of resources are necessary for their creation.

Just like a quantum state, $\Upsilon_{n+1}$ is a positive matrix. However, in contrast to quantum states, $\Upsilon_{n+1}$ has to encapsulate the causal ordering of sequential measurements [19, 39]. Specifically, the structure of $\Upsilon_{n+1}$ has to be such that the choice of instrument at time $t_j$ cannot influence the statistics observed at any earlier time $t_i < t_j$. This requirement imposes a hierarchy of trace conditions [19]:

$$\begin{aligned}
\text{tr}_{n+1}^{\text{i}}(\Upsilon_{n+1}) &= \mathbb{1}_n^{\text{o}} \otimes \Upsilon_n \\
\text{tr}_n^{\text{i}}(\Upsilon_n) &= \mathbb{1}_{n-1}^{\text{o}} \otimes \Upsilon_{n-1} \\
&\vdots \\
\text{tr}_2^{\text{i}}(\Upsilon_2) &= \mathbb{1}_1^{\text{o}} \otimes \Upsilon_1,
\end{aligned} \quad (4)$$

where $\Upsilon_1 \in \mathcal{B}(\mathcal{H}_1^{\mathrm{i}})$ is a quantum state[2]. These equations also fix the overall trace of $\Upsilon_{n+1}$ to be $\mathrm{tr}(\Upsilon_{n+1}) = d_n^{\mathrm{o}} \cdot d_{n-1}^{\mathrm{o}} \cdots d_1^{\mathrm{o}} := d^{\mathrm{o}}$, with $d_x^{\mathrm{o}} = \dim(\mathcal{H}_x^{\mathrm{o}})$. Vice versa, *any* positive matrix that satisfies the above trace conditions can be considered the Choi matrix of an underlying quantum stochastic process, or, equivalently, of an underlying quantum causal model [19].

To see why the above conditions ensure causal ordering, consider, for example, the case of three times $\{t_1, t_2, t_3\}$. Statistics at time $t_1$ should not depend on the choice of instruments $\mathcal{J}_2$ and $\mathcal{J}_3$ at times $t_2$ and $t_3$. Any given choice of these latter two instruments implies that – on average – at times $t_2$ and $t_3$, the experimenter performs CPTP maps with Choi matrices $M_2$ and $M_3$ respectively. As the output space of $\mathcal{J}_3$ is trivial, it is a POVM, implying $M_3 = \sum_{x_3} M_{x_3} = \mathbb{1}_3^{\mathrm{i}}$. With this, we see that

$$\sum_{x_2 x_3} \mathbb{P}(x_3, x_2, x_1 | \mathcal{J}_3, \mathcal{J}_2, \mathcal{J}_1) = \mathrm{tr}[(M_3^{\mathrm{T}} \otimes M_2^{\mathrm{T}} \otimes M_{x_1}^{\mathrm{T}}) \Upsilon_3] = \mathrm{tr}[(M_2^{\mathrm{T}} \otimes M_{x_1}^{\mathrm{T}} (\mathbb{1}_2^{\mathrm{o}} \otimes \Upsilon_2)]$$

$$= \mathrm{tr}[M_{x_1}^{\mathrm{T}} (\mathbb{1}_1^{\mathrm{o}} \otimes \Upsilon_1)] = \mathbb{P}(x_1 | \mathcal{J}_1), \tag{5}$$

where we have alternatingly used the property $\mathrm{tr}_j^{\mathrm{o}}(M_j) = \mathbb{1}_j^{\mathrm{i}}$ of CPTP maps and the causality conditions of Eqs. (4). As $\Upsilon_1$ is independent of the choice of $\mathcal{J}_3$ and $\mathcal{J}_2$, so is $\mathbb{P}(x_1 | \mathcal{J}_3, \mathcal{J}_2, \mathcal{J}_1)$.

In order to investigate the structural properties of combs and to see how they stem from the underlying dynamical 'building blocks' (i.e., the initial state $\rho_{AR}$, as well as the intermediate maps $\{\mathcal{L}_j\}$), it is convenient to introduce the *link product* [19]. For example, $\Upsilon_{n+1}$ can be straightforwardly computed as

$$\Upsilon_{n+1} = \rho_{AR} \star L_1 \star \cdots \star L_n, \tag{6}$$

where the link product $\star$ between two matrices $F \in \mathcal{B}(\mathcal{H}_x \otimes \mathcal{H}_y)$ and $G \in \mathcal{B}(\mathcal{H}_y \otimes \mathcal{H}_z)$ is given by

$$F \star G = \mathrm{tr}_y[(F \otimes \mathbb{1}_z)(G^{\mathrm{T}_y} \otimes \mathbb{1}_x)]. \tag{7}$$

Put shortly, the link product traces two Choi matrices over the spaces they share, and corresponds to a tensor product on the remaining spaces (importantly, $F \star G = F \otimes G$ if $F$ and $G$ are defined on disjoint spaces). Intuitively, "$\star$" expresses the concatenation "$\circ$" of maps to the case of Choi matrices, i.e., the Choi matrix of $\mathcal{F} \circ \mathcal{G}$ is given by $F \star G$. For example, the action of a map $\mathcal{L}$ on a state $\rho$ can be written as $\mathcal{L}[\rho] = L \star \rho$. Importantly, the link product satisfies $F \star (G \star H) = (F \star G) \star H = F \star G \star H$, it is commutative for all cases we consider, and the link product of positive matrices is itself a positive matrix. To keep better track of the involved spaces, we will often additionally label Choi matrices with the spaces they are defined on. While we provide a more detailed discussion of the link product in App. A (see [19] for thorough derivations), the above definition is sufficient for our purposes. We will make use of it frequently in what follows to derive the properties of combs from their underlying building blocks.

Mapping the somewhat abstract linear functional $\mathcal{T}_{n+1}$ onto its Choi matrix $\Upsilon_{n+1}$ has the advantage that the latter is – up to normalization – a quantum state, and all temporal correlations that the given process $\mathcal{T}_{n+1}$ can display are now encoded in the spatial correlations of the (unnormalized) quantum state $\Upsilon_{n+1}$. Consequently, the vast machinery that has been developed for the analysis of bi- and multipartite entanglement in quantum states [10, 53, 54] can be used to analyse temporal correlations that are genuinely quantum. Such a program has recently led to the definition and investigation of genuine quantum memory in temporal processes [32]. Naturally, as combs are not normalized to unity, in what follows, when we

---

[2]Note that the role of input and output spaces for the comb and the CP maps it acts on is interchanged (outputs of the CP maps are inputs of the comb, and vice versa), such that all the above trace conditions are with respect to spaces that are labelled by i, not o.

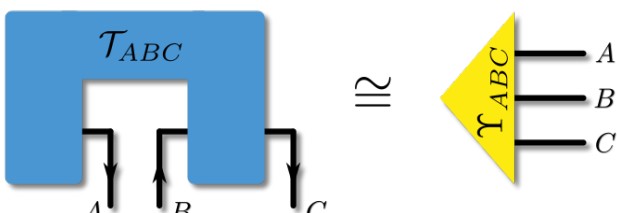

Figure 2: **Two-step process.** A two step process can be considered (up to normalization) a three-partite quantum state, where each of the parties has a different role. Alice (A) can measure the state of the system at time $t_1$, Bob (B) can feed forward a state at $t_1$, and Charlie (C) can measure the state of the system at $t_2$. To emphasize the causal ordering of the process, arrows have been added to the 'legs' of the comb.

speak of entanglement, we will always understand it *up to normalization*; then, for example, a comb $\Upsilon_{ABC}$ is separable in the splitting $A : BC$, if it can be written in the form $\sum_\alpha \rho_A^{(\alpha)} \otimes D_{BC}^{(\alpha)}$, where $\rho_A^{(\alpha)}, D_{BC}^{(\alpha)} \geq 0$ for all $\alpha$.

Throughout, we will predominantly analyse the three-party case, both investigating the types of genuinely tripartite entanglement in temporal processes that can persist, as well as the necessary and sufficient conditions on the underlying dynamics for their occurrence. This case is simple enough to allow for explicit results, yet already displays genuine quantum effects [32]. To simplify notation, we denote the involved Hilbert spaces by $\mathcal{H}_A, \mathcal{H}_B$, and $\mathcal{H}_C$ instead of $\mathcal{H}_{1^i}, \mathcal{H}_{1^o}$, and $\mathcal{H}_{2^i}$, and the corresponding comb by $\Upsilon_{ABC}$. Consequently, the combs we will consider satisfy

$$\Upsilon_{ABC} \geq 0, \quad \mathrm{tr}_C(\Upsilon_{ABC}) = \mathbb{1}_B \otimes \Upsilon_A, \quad \mathrm{tr}(\Upsilon_A) = 1. \tag{8}$$

Besides notational simplification, this relabelling has the additional advantage of providing an intuitive role that each of the 'parties' play. $A$ corresponds to Alice measuring the system state at $t_1$, $B$ corresponds to Bob feeding forward a state at $t_1$ (*after* Alice's measurement), and $C$ corresponds to Charlie measuring the final state of the system at time $t_2$ (see Fig. 2). With this, for example, different kinds of bipartite entanglement in $\Upsilon_{ABC}$ correspond to different kinds of 'control' that each party has over the correlations the other two share. Throughout this article, we will adopt the convention that the Choi states of maps are labeled in the temporal order in which the respective spaces appear.

## 2.2 Preliminaries: Entanglement

While structurally similar to *spatial* quantum correlations, it is a priori unclear how the additional causality constraints of Eqs. (4) affect the phenomena of multipartite entanglement that can persist in the temporal setting. Before studying this in detail we will first set the ground and review some important definitions and results from entanglement theory.

To begin with, states are entangled if they cannot be prepared with local operations (by potentially spatially separated parties) and classical communication among them (LOCC). In the bipartite case this implies that they cannot be written of the form

$$\rho_{A:B}^{\mathrm{sep}} = \sum_i p_i \rho_i^A \otimes \rho_i^B, \tag{9}$$

where here and in the following $\{p_i\}$ is a probability distribution and $\rho_i^A$ ($\rho_i^B$) are valid density matrices of party A (party B) respectively. If states are of this form they are called

separable, and we will denote them by $\rho_{A:B}^{\text{sep}}$. An important example of a bipartite entangled state which is in fact maximally entangled and which plays also an important role in the CJI is $|\Phi^+\rangle = 1/\sqrt{2}(|00\rangle + |11\rangle)$. Whether the Choi matrix is entangled or not provides information about the underlying channel. More precisely, the Choi matrix is separable if and only if the corresponding quantum channel is entanglement-breaking (see [55] and references therein).

In the multipartite scenario the situation is more complex (see, e.g. [53]). The straightforward generalization of Eq. (9) yields fully separable states. However, it may also be that a multipartite state of $N$ parties contains at most entanglement among $k$ parties for $k = 2, \ldots, N$. For the case that the state indeed contains $N$-partite entanglement (in any possible decomposition) it is called genuine multipartite entangled (GME). As mentioned before, we will mainly be interested in the three-party case. In this case if a state is entangled one may either observe only bipartite entanglement or genuine tripartite entanglement. It may be for example that a state is separable with respect to some specific bipartition, e.g. $A : BC$. That is, it can be written as $\rho_{A:BC}^{\text{sep}} = \sum_i p_i \rho_i^A \otimes \rho_i^{BC}$ and it does not contain any entanglement between party $A$ and parties $B, C$ which we denote by $\mathcal{E}(A : BC) = 0$. Note that, however, in this case parties $B$ and $C$ can still share entanglement (in contrast to a fully separable state). Any biseparable state is then some mixture of biseparable states with respect to different bipartitions, i.e., it can be written in the form

$$\rho = q_1 \rho_{A:BC}^{\text{sep}} + q_2 \rho_{B:AC}^{\text{sep}} + q_3 \rho_{C:AB}^{\text{sep}}, \tag{10}$$

where $\{q_i\}$ are probabilities that sum to unity. Any state that is not of this form is genuinely tripartite entangled.

By definition any GME state has to contain bipartite entanglement across all cuts, i.e., $\mathcal{E}(A : BC) > 0$, $\mathcal{E}(B : AC) > 0$ and $\mathcal{E}(C : AB) > 0$. This does not necessarily hold true if one particle gets lost, i.e., the marginals may be separable or, stated differently, there exist GME states for which after performing a partial trace over $C$ the resulting state is separable, i.e., $\mathcal{E}(A : B) = 0$, and analogously for all other parties. For example the GHZ state, $|GHZ\rangle = (|000\rangle + |111\rangle)/\sqrt{2}$ shows this property. Performing a partial trace is a very special case of a POVM measurement. As has been also shown one can find GME states for which any possible POVM measurement yields a separable state on the remaining parties [33]. Assuming that for example party $C$ is performing the measurement, we will denote this conditional form of entanglement by either $\mathcal{E}(A : B|c) > 0$ in case there exists some post-measurement state on $A$ and $B$ which is entangled or $\mathcal{E}(A : B|c) = 0$ otherwise. Note that in the first case the measurements of party $C$ may be able to influence the resource $A$ and $B$ share. We will consider the implications of such an effect on temporal processes in Sec. 3.3. Note further that the notion of conditional entanglement considered here is different from localizable entanglement [56], which is defined as the maximal entanglement among two parties that can be obtained on average via local measurements on the other parties.

There exist several criteria which may allow one to detect that a state is entangled. In particular, for the two-qubit case and the case of a qubit and a qutrit a necessary and sufficient condition for entanglement which can be easily evaluated exists. It is known as the PPT criterion or Peres-Horodecki criterion [57, 58] and states the following: a state with total dimension $d = d_A d_B \leq 6$ is separable if and only if its partial transposition yields a positive semi-definite operator, i.e., it has a positive partial transpose (PPT). For higher dimensions any separable state is PPT however the converse is not true. Hence, the PPT criterion still constitutes a necessary criterion for separability and still allows to certify entanglement of bipartite states.

Entanglement can also be detected by considering entanglement witnesses (EWs) [58, 59]. These are operators which yield a non-negative expectation value for all separable states but which detect at least one entangled state via a negative expectation value. For any entangled

state (independent of the number of parties or local dimensions) there exists an EW that heralds the presence of entanglement [58]. It is, however, not clear how to construct it in general.

Matters are significantly simplified, if the set of separable states is relaxed to the set of states that are PPT mixtures, which are written as

$$\rho = q_1 \rho_{A:BC}^{\text{ppt}} + q_2 \rho_{B:AC}^{\text{ppt}} + q_3 \rho_{C:AB}^{\text{ppt}}, \tag{11}$$

where each term in the above equation corresponds to a state with a PPT in the respective splitting. Any state that cannot be written in the form of Eq. (11) is GME, but there are GME states that are PPT mixtures. Importantly, using the concept of EWs, membership to the set of PPT mixtures can be decided by means of the following semi-definite program (SDP) [60]

$$\begin{aligned} \min \quad & \text{tr}(W\rho) \\ \text{such that} \quad & W = Q_M + P_M^{\text{T}_M} \text{ with } \text{tr}(W) = 1, \\ & P_M \geq 0, \text{ and } Q_M \geq 0 \ \forall M \in \{A, B, C\}. \end{aligned} \tag{12}$$

This can be easily seen as for any state $\rho$ of the form in Eq. (10) it holds that

$$\begin{aligned} \text{tr}(W\rho) = & q_1 \text{tr}(Q_A \rho_{A:BC}^{\text{sep}}) + q_1 \text{tr}(P_A [\rho_{A:BC}^{\text{sep}}]^{\text{T}_A}) \\ & + q_2 \text{tr}(Q_B \rho_{B:AC}^{\text{sep}}) + q_2 \text{tr}(P_B [\rho_{B:AC}^{\text{sep}}]^{\text{T}_B}) \\ & + q_3 \text{tr}(Q_C \rho_{C:AB}^{\text{sep}}) + q_3 \text{tr}(P_C [\rho_{C:AB}^{\text{sep}}]^{\text{T}_C}) \geq 0. \end{aligned} \tag{13}$$

Here we used that $\text{tr}(\sigma P_M^{\text{T}_M}) = \text{tr}(\sigma^{\text{T}_M} P_M)$ and the inequality follows from the fact that $[\rho_{A:BC}^{\text{sep}}]^{\text{T}_A} \geq 0$ (and analogously for the other splittings) and by definition $P_M, Q_M \geq 0$. Hence, in case a negative value is observed the state is certified to be GME. Moreover, it has been shown that for any state that is GME there exists a witness of the form given in the SDP (12) which detects it [60].

For certain classes of states the following analytical criterion is particularly useful to prove GME. It has been shown [61] that for a biseparable $n$-qubit state, $\rho^{(n)}$, it holds that

$$|\rho_{(0...0,1...1)}^{(n)}| \leq \frac{1}{2} \sum_{|I| \in \{1,...,n-1\}} \sqrt{\rho_{(I,I)}^{(n)} \rho_{(\bar{I},\bar{I})}^{(n)}}, \tag{14}$$

where here we use the notation $\rho_{(ijk...,\alpha\beta\gamma...)}^{(n)} = \langle ijk... | \rho^{(n)} | \alpha\beta\gamma... \rangle$, $I = (i_1, i_2 ..., i_n)$ with $i_j \in \{0, 1\}$ and $\bar{I}$ is obtained from the tuple $I$ by swapping zeroes and ones. Moreover, we denote by $|I|$ the Hamming weight of the tuple and $\sum_{|I|=k}$ denotes a sum over all tuples which have Hamming weight $k$. In case the inequality is violated the state is GME.

So far we considered the question among how many parties entanglement has to persist in order to create a state. However, one may also be interested in organizing states into different classes of entanglement. Such a classification can be achieved by considering stochastic local operations assisted by classical communication (SLOCC). If one can transform with non-vanishing probability the pure state $|\Psi\rangle$ into another pure state $|\Phi\rangle$ via LOCC and the reverse transformation from $|\Phi\rangle$ to $|\Psi\rangle$ is also possible via SLOCC, both states are within the same SLOCC class [62]. Mathematically, this corresponds to $|\Phi\rangle \propto A_1 \otimes ... \otimes A_n |\Psi\rangle$ with $A_i$ being local invertible operators. For three qubits there exist two different SLOCC classes which are genuine tripartite entangled [62], the W- class and the GHZ-class. Well-known representatives of these classes are the W state, $|W\rangle \propto |001\rangle + |010\rangle + |100\rangle$, and the GHZ state, $|GHZ\rangle \propto |000\rangle + |111\rangle$.

The concept of SLOCC classes can be generalized from pure states to mixed states by defining such a class as the convex hull over all pure states within the closure of a SLOCC

class [63]. As an example, consider the W-class whose closure also includes all biseparable and fully separable states. Hence, the W-class (for mixed states) contains all states that are convex combinations of pure states within the W-class, biseparable and fully separable states. As the closure of the GHZ class for three qubits contains all pure three-qubit states, all states are contained in the GHZ class for mixed states. Moreover, all states which are in GHZ\ W have the property that in any decomposition at least one state is contained in the GHZ-class (of pure states). Such states can also be identified by using witness operators [63, 64]. An example of such a SLOCC witness is given by [63] $W = (3/4)\mathbb{1} - |GHZ\rangle \langle GHZ|$, which gives a positive expectation value for any state within the W-class but is able to detect, e.g., the GHZ state. Another way to distinguish among different SLOCC classes can be by considering SL invariants [62, 65]. The tangle [66] (which is also an entanglement measure) is such a quantity, it is non-zero for states within GHZ\W and zero for all states in the W-class [62]. Below, we will use these criteria to show that processes of all SLOCC classes exist for the case of two times, i.e., three parties.

## 2.3   Preliminaries: States vs. Processes

As mentioned, via the CJI, all quantum combs can be considered as quantum states (up to normalization). Hence, the concepts and tools presented above can be also applied for their characterization. When doing so we will use the normalization for combs, i.e., $\text{tr}(\Upsilon_{ABC}) = d_B$ and if necessary modify the criteria accordingly. We emphasize that the causality constraints prohibits that *all* states can be considered as temporal processes. This already holds true for the case of two times $t_1, t_2$, i.e., the tripartite case with the involved Hilbert spaces $\mathcal{H}_A, \mathcal{H}_B$, and $\mathcal{H}_C$. Here, for example, the GHZ state is a genuinely tripartite entangled quantum state, but *not* (proportional to) a proper comb $\Upsilon_{ABC}$. More generally, it is straightforward to see that any process that is described by a *pure* comb cannot be genuinely multipartite entangled, as it has to be of the form $\Upsilon_{n+1} = |\Phi_s\rangle\langle\Phi_s| \otimes V_1 \otimes \cdots \otimes V_n$, where $\{V_1, \ldots, V_n\}$ are the (pure) Choi matrices of unitary maps that act on the system alone.

This structural difference between states and processes also extends to the respective physical implications of entanglement in the spatial and the temporal setting. While for quantum states, entanglement is a statement about correlations between spacelike separated measurements, the correlations in a comb can be given a direct operational meaning in terms of causal structure of the underlying process. Discerning and determining different causal influences is the object in the field of (quantum) causal modelling [1, 2]. For two parties, correlations between them can exist due to a *common cause*, i.e., they are correlated, but cannot causally influence each other; or a *direct cause* between them, i.e., one party's actions can influence the statistics of the other.

Expressed in quantum mechanical terms, a common cause (between, say, parties $A$ and $C$) corresponds to the two parties sharing a quantum state, but no channel between them to communicate. As the state can be correlated, their measurement outcomes when probing said state can also be correlated, but they cannot influence each other. When the state that is shared is entangled, we will call it a *quantum* common cause.

On the other hand, a direct cause, say, between $B$ and $C$, implies that information can be sent from $B$ to $C$ (or vice versa), i.e., they share a communication channel between them. In this case, correlations between the two parties stem from a direct causal influence. We will call a direct cause *quantum*, if the channel that is shared admits the sending of quantum information, i.e., if it is not entanglement breaking.

For a process with three parties of the form discussed above (see Fig. 2), both of these types of correlations can be read off directly from the comb $\Upsilon_{ABC}$. As Fig. 2 suggests, $B$ is the only party that can share direct cause correlations with the other two (Bob is the only party that can feed something into the process), while $A$ and $C$ can only share common cause

correlations. More precisely, we will both consider the correlations across different splits in the full comb $\Upsilon_{ABC}$ as well as its (conditional) marginals. Then, depending on the respective comb, the considered Choi state correspond to a channel between parties – i.e., a direct cause – or a state shared by different parties – i.e., a common cause.

The type of correlation then tells us directly, if the cause is quantum or not. For example, if $\Upsilon_{AC}$ is entangled, then Alice and Charlie share an entangled state, implying that they have a quantum common cause. On the other hand, if the considered Choi state is that of a channel, then entanglement in a cut means that the underlying channel is not entanglement breaking [67]. Consequently, if, say, $\Upsilon_{BC}$ is entangled, then there is a quantum direct cause between Bob and Charlie. Any study of quantum correlations in combs has thus a direct operational implication for the process at hand that is investigated.

Naturally, beyond the two-party paradigm, the landscape of causal structures becomes more complex. For example, as mentioned it has been shown that, in quantum mechanics, common cause and direct cause scenarios can be superposed [30, 31]. More generally, combs can display genuinely multipartite entanglement. While somewhat harder to interpret in terms of common and direct causes, following the definition of Choi states, a GME comb has nonetheless an operational interpretation: the corresponding process can be harnessed to create GME from a collection of mutually uncorrelated maximally (bipartite) entangled states by, respectively, acting on only one of their parts.

Here, we will investigate in detail the entanglement structure of processes on two times and beyond; i.e., we investigate what types of entanglement, both bi- and multipartite, can exist. This, in turn, answers the question of what genuinely quantum phenomena processes in time can display. Additionally, we will study the basal building blocks required for the implementation of such different types of processes, providing a comprehensive characterization of the resources required for to exploit such genuine quantum features. Both of these results, in turn, provide insights into the multilayered nature of causal relations that persist in quantum processes.

## 3 Bipartite entanglement in combs – necessary conditions

As a first step, before considering genuine multipartite entanglement in quantum processes, we consider the simpler case of bipartite entanglement, and answer the questions what kinds of entanglement are possible, given the causality requirements of Eq. (8), and what underlying resources are necessary for its creation. Put in terms of quantum causal modelling, entanglement of a comb in different splittings amounts to different types of quantum causal relations [30, 31]. Their analysis both allows one to make inferences about the necessary resources required to create different types of entanglement in combs, and prepares the discussion of the genuinely tripartite case.

The bipartite case can be considered in two different ways. On the one hand, assuming a 'global' position, entanglement in the splittings

$$\{\mathcal{E}(A:BC),\ \mathcal{E}(B:AC),\ \mathcal{E}(C:AB)\} \tag{15}$$

is of interest. On the other hand, considering each leg in the comb in Fig. 2 as a party that either wants to communicate with another party, and/or tries to influence the communication of the other two, entanglement of the form

$$\{\mathcal{E}(A:B|c),\ \mathcal{E}(A:C|b),\ \mathcal{E}(B:C|a)\} \tag{16}$$

is relevant, where conditioning implies that the respective entanglement can depend on a measurement or preparation that the third party performed. Evidently, entanglement of the

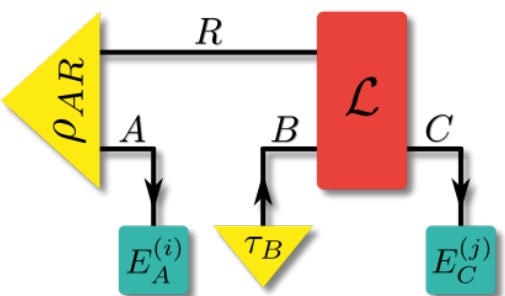

Figure 3: **Building blocks of a two-step process.** The structure of a two-step process depends on the respective properties of the initial system-environment state $\rho_{AR} \in \mathcal{B}(\mathcal{H}_A \otimes \mathcal{H}_R)$, the map $\mathcal{L} : \mathcal{B}(\mathcal{H}_A \otimes \mathcal{H}_R) \rightarrow \mathcal{B}(\mathcal{H}_C)$, and their interplay. Identifying each of the 'legs' of the process with a different party, Alice and Charlie can measure their respective inputs (with corresponding POVM elements $E_A^{(i)}$ and $E_C^{(j)}$), while Bob can feed states $\tau_B$ into the process.

latter case implies entanglement in the former, global case, as no local operation can create entanglement. Here, we first investigate the necessary conditions for entanglement in the conditional case, while the unconditional one will be discussed in detail in the next section.

### 3.1 Quantum common cause – Entanglement $\mathcal{E}(A : C|b) > 0$

Depending on the interplay of the underlying building blocks of the process, Alice and Charlie can – potentially conditioned on the state that Bob inserts into the process – share entanglement. Expressed in the language of causal modelling [1,2], such a scenario would (possibly depending on the state that Bob feeds forward) imply a *quantum common cause* [3,30] between Alice and Charlie.

Evidently, for this type of entanglement to persist, the initial system-environment needs to be entangled between the system $A$ and the environment $R$. Additionally, the subsequent system-environment map $\mathcal{L} : \mathcal{B}(\mathcal{H}_B \otimes \mathcal{H}_R) \rightarrow \mathcal{B}(\mathcal{H}_C)$ (with corresponding Choi matrix $L_{BRC}$ cannot destroy this entanglement (see Fig. 3 for a graphical depiction of the considered process). These two requirements can be summarized in the following Proposition:

**Proposition 1.** *If a process $\Upsilon_{ABC}$ satisfies $\mathcal{E}(A : C|b) > 0$ for some state $\tau_B$ prepared by Bob, then the initial system-environment state $\rho_{AR}$ is entangled, and there exists a pure state $|\Phi\rangle\langle\Phi|_B \in \mathcal{B}(\mathcal{H}_B)$ such that $|\Phi\rangle\langle\Phi|_B \star L_{BRC} \in \mathcal{B}(\mathcal{H}_R \otimes \mathcal{H}_C)$ is proportional to an entangled state (i.e., the corresponding map $\widetilde{\mathcal{L}} : \mathcal{B}(\mathcal{H}_R) \rightarrow \mathcal{B}(\mathcal{H}_C)$ is not entanglement breaking).*

*Proof.* The overall process $\Upsilon_{ABC}$ can be computed directly from its building blocks as

$$\Upsilon_{ABC} = \rho_{AR} \star L_{BRC} = \mathrm{tr}_R(\rho_{AR} L_{BRC}^{\mathrm{T}_R}), \tag{17}$$

where we have omitted the respective identity matrices. Now, if $\rho_{AR}$ is separable, it can be decomposed as $\rho_{AR} = \sum_i p_i \rho_A^{(i)} \otimes \eta_R^{(i)}$, which leads to an overall comb of the form

$$\Upsilon_{ABC} = \sum_i p_i \rho_A^{(i)} \otimes \mathrm{tr}_R(\eta_R^{(i)} L_{BRC}^{\mathrm{T}_R}), \tag{18}$$

which is – up to normalization – a quantum state that is separable in the splitting $A : C$, independent of what Bob does.

On the other hand, assuming that $|\Phi\rangle\langle\Phi|_B \star L_{BRC}$ is an entanglement breaking channel for all states $|\Phi\rangle_B$ implies that $\tau_B \star L_{BRC}$ is entanglement breaking for any state $\tau_B$ that Bob feeds

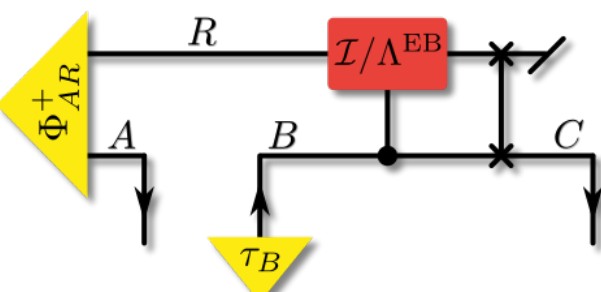

Figure 4: **Bob controls the entanglement between Alice and Charlie.** Controlled on Bob's input, no transformation (i.e., $\mathcal{I}$) takes place on the environment line, or an entanglement breaking map $\Lambda_{\text{EB}}$ is performed. Afterwards, system and environment are swapped and the environment is discarded.

into the process. As such, we have $\tau_B \star L_{BRC} = \sum_\alpha E_R^{(\alpha)} \otimes \xi_C^{(\alpha)}$ [55, 67], where $\{E_R^{(\alpha)}\}$ is a POVM and $\{\xi_C^{(\alpha)}\}$ are quantum states. Then, one obtains for the resulting comb shared between Alice and Charlie:

$$\Upsilon_{AC|b} = \Upsilon_{ABC} \star \tau_B = \sum_\alpha \rho_{AR} \star \tau_B \star L_{BRC} = \sum_\alpha \text{tr}_R(\rho_{AR}^{\text{T}_R} E_R^{(\alpha)}) \otimes \xi_C^{(\alpha)}, \tag{19}$$

which is a separable state on $\mathcal{B}(\mathcal{H}_A \otimes \mathcal{H}_C)$. □

A simple example for a process that satisfies $\mathcal{E}(A : C|b) > 0$ is one where the initial system-environment is a maximally entangled state, and the action of the map $\mathcal{L}$ is to swap the system and the environment, and subsequently discard the environment. In this case, the resulting state shared between Alice and Charlie would be the maximally entangled state $\Phi_{AC}^+$, independent of what state Bob feeds into the process.

More generally, it is also possible that Bob can actively 'switch' the entanglement between Alice and Charlie on or off by choosing his input state $\tau_B$ appropriately. To see this, consider a variation of the above channel, where, before the swap, depending on the input state of Bob, either an identity channel $\mathcal{I}$ or an entanglement breaking channel $\Lambda^{\text{EB}}$ is implemented (see Fig 4). Concretely, let the action of this map on a $BR$ state $\rho_{BR}$ be given by

$$\rho_{BR} \mapsto |0_B\rangle\langle 0_B| \otimes \langle 0_B|\rho_{BR}|0_B\rangle + |1_B\rangle\langle 1_B| \otimes \Lambda^{\text{EB}}[\langle 1_B|\rho_{BR}|1_B\rangle]. \tag{20}$$

In this case, if Bob feeds $\tau_B = |0_B\rangle\langle 0_B|$ into the process, then the resulting $AC$ state is equal to the maximally entangled state $\Phi_{AC}^+$, while, if he feeds forward $\tau_B = |1_B\rangle\langle 1_B|$ into the process, then Alice and Bob share the separable state $\Lambda^{\text{EB}} \otimes \mathcal{I}_A[\Phi_{AC}^+]$.

The above discussion also provides a direct intuitive interpretation of $\mathcal{E}(A : C|b) > 0$ for a comb $\Upsilon_{ABC}$; it implies that (possibly depending on Bob's input), Alice (i.e., an experimenter at time $t_1$) and Charlie (i.e., an experimenter at time $t_2$) share an entangled state, or, equivalently, there is a quantum common cause between Alice and Charlie, depending on what Bob does. Importantly – unlike in the two cases that follow – Bob can control the entanglement that is shared between Alice and Charlie *deterministically*, i.e., no conditioning on measurement outcomes is necessary. Phrased with respect to the above example, Bob can decide beforehand, which of the possible states he wants Alice and Charlie to share. While there are infinitely many deterministic preparations (all states $\tau_B$ that Bob can feed into the process), there is only one deterministic POVM element [68] (discarding the system). Consequently, Alice and Charlie can control the causal correlations of the respective other two parties, but will only know what particular correlations they controlled on *after* their measurement.

### 3.2 Quantum direct cause – Entanglement $\mathcal{E}(B : C|a)$

While $\mathcal{E}(A : C|b)$ concerns the entanglement between two output legs, i.e., two parts of a quantum state, $\mathcal{E}(B : C|a)$ concerns the entanglement between an input leg ($B$) and an output leg ($C$). Consequently, the property $\mathcal{E}(B : C|a) > 0$ is directly related to the entanglement preserving properties of the map $\mathcal{L} : \mathcal{B}(\mathcal{H}_B \otimes \mathcal{H}_R) \to \mathcal{B}(\mathcal{H}_C)$ from time $t_1$ to time $t_2$. The presence (possibly conditioned on Alice's outcome) of this type of entanglement can thus be considered a *quantum* direct cause between Bob and Charlie. In particular, we have the following Proposition:

**Proposition 2.** *If a process $\Upsilon_{ABC}$ satisfies $\mathcal{E}(B : C|a) > 0$ for some POVM element $E_A$ in Alice's lab, then there exists a pure state $|\Phi_R\rangle\langle\Phi_R|$ on the environment, such that $|\Phi_R\rangle\langle\Phi_R| \star L_{BRC} \in \mathcal{B}(\mathcal{H}_B \otimes \mathcal{H}_C)$ is proportional to an entangled state (i.e., the corresponding map $\bar{\mathcal{L}} : \mathcal{B}(\mathcal{H}_B) \to \mathcal{B}(\mathcal{H}_C)$ is not entanglement breaking).*

*Proof.* Let the initial system-environment state be $\rho_{AR}$. If Alice performs a measurement with an outcome corresponding to the POVM element $E_A$, the remaining comb shared by Bob and Charlie is given by $E_A^{\mathrm{T}} \star \Upsilon_{ABC}$ (where the transpose on $E_A$ is necessary to comply with the definition of the link product) which, written in terms of its building blocks is equal to

$$\Upsilon_{BC|a} = E_A^{\mathrm{T}} \star \rho_{AR} \star L_{BRC}. \tag{21}$$

Up to normalization, $E_A^{\mathrm{T}} \star \rho_{AR}$ is a quantum state on $\mathcal{B}(\mathcal{H}_R)$. If there exists no pure state $|\Phi_R\rangle\langle\Phi_R|$ such that $|\Phi_R\rangle\langle\Phi_R| \star L_{BRC}$ is entangled, then $\Upsilon_{BC|a}$ in Eq. (21) cannot be proportional to an entangled state. $\qquad\square$

As $\Upsilon_{BC|a}$ is (proportional to) the Choi matrix of a quantum channel from Bob to Charlie (where the proportionality constant is equal to the probability for Alice to measure the outcome corresponding to $E_A$), $\mathcal{E}(B : C|a) > 0$ implies that for some measurement outcome in Alice's laboratory, Bob has an entanglement preserving channel to Charlie at his disposal, i.e., he can send him quantum information. Unlike in the previous case, conditioning on an outcome in Alice's laboratory is in general not deterministic, and the probability for the POVM element $E_A$ to occur is given by $\mathrm{tr}(E_A \rho_{AR})$. As already mentioned, the only deterministic POVM element that Alice can perform is $\mathbb{1}_A$, which amounts to discarding her part of the initial state $\rho_{AR}$.

It is straightforward to find examples of processes that satisfy $\mathcal{E}(B : C|a) > 0$ for all POVM elements $E_A$ (including $E_A = \mathbb{1}_A$). One such example is a process without environment, with a pure state initial state $|\Psi_A\rangle$ and an identity map between Bob and Charlie, which yields the valid comb $\Upsilon_{ABC} = |\Psi_A\rangle\langle\Psi_A| \otimes \widetilde{\Phi}^+_{BC}$, where the unnormalized maximally entangled state $\widetilde{\Phi}^+_{BC}$ is the Choi matrix of the identity channel $\mathcal{I}_{B \to C}$ (see App. A). On the other hand, there are processes that yield an entanglement preserving channel between Bob and Charlie for each of the outcomes of Alice (for a given POVM), but do display an entanglement breaking channel if no conditioning takes place.

For example, let all systems (including the environment $R$) be qubits, the initial system-environment state a maximally entangled state $\Phi^+_{AR}$, the map $\mathcal{L}$ be a controlled Pauli-$Z$ gate, with control on the environment, and Alice performs a measurement in the computational basis (see Fig. 5). Then, for outcome 0 of Alice's measurement (which occurs with probability $1/2$), the corresponding map between $B$ and $C$ is the identity map with corresponding Choi matrix $\widetilde{\Phi}^+_{BC}$, while for the outcome 1 the corresponding map between $B$ and $C$ is given by the Pauli-$Z$ gate (with corresponding Choi matrix $\widetilde{\Phi}^-_{BC}$). Each of these maps is entanglement preserving, however, the average map (with corresponding Choi matrix $\widetilde{\Phi}^+_{BC} + \widetilde{\Phi}^-_{BC} = |00\rangle\langle00|_{BC} + |11\rangle\langle11|_{BC}$) is the completely dephasing map, which is entanglement breaking. Consequently, for a given process $\Upsilon_{ABC}$, Alice might be able to switch on and off Bob's capability to transmit quantum information to Charlie, but she cannot do so deterministically.

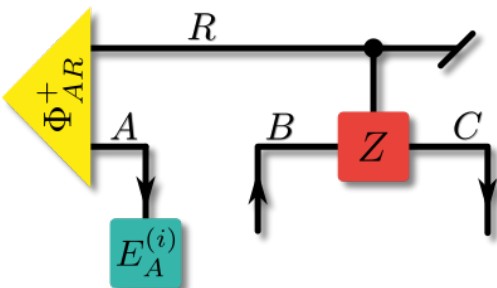

Figure 5: **Alice controls the entanglement between Bob and Charlie.** Alice measures in the computational basis. If she obtains outcome 0, the channel between $B$ and $C$ is the identity channel. Otherwise, it is given by $\rho \mapsto \sigma_z \rho \sigma_z$, where $\sigma_z$ is the $Z$-Pauli matrix. Both of these channels are unitary – and hence their Choi matrices are proportional to entangled states – but on average, i.e., if Alice simply discards her system, the channel between $B$ and $C$ is the completely dephasing one, which is entanglement breaking.

### 3.3 Channels from the future to the past – Entanglement $\mathcal{E}(A:B|c)$

So far, the causality constraints on $\Upsilon_{ABC}$ only played a minor role in the considerations of possible entanglement in different splittings. For example, as we have seen above, entanglement in the splitting $B:C$ can exist both deterministically (for the case that Alice's POVM element is $\mathbb{1}_A$), as well as probabilistically (for $E_A \neq \mathbb{1}_A$). This freedom no longer exists if the entanglement between Alice and Bob is considered. From the causality constraint (8), we have $\mathrm{tr}_C(\Upsilon_{ABC}) = \mathbb{1}_B \otimes \Upsilon_A$, implying that if Charlie discards the state he receives, then there are no correlations (quantum or classical) between Bob and Alice. Otherwise, the choice of Bob's input could influence the state that Alice receives, which is forbidden by causality. Consequently, $\mathcal{E}(A:B|c) > 0$ means that, by conditioning on a measurement outcome in Charlie's lab (corresponding to the POVM element $E_C$), quantum information can be sent from Bob to Alice (i.e., 'from the future to the past' [69]). Such simulation scenarios by means of conditioning, have been discussed in different contexts in the literature [27, 69–73]. For this to be possible, the initial system-environment state $\rho_{AR}$ has to be entangled, and the map $\mathcal{L}$ has to be able to entangle $R$ and $B$. More precisely, we have the following Proposition:

**Proposition 3.** *If a process $\Upsilon_{ABC}$ satisfies $\mathcal{E}(A:B|c) > 0$, then the initial system-environment state $\rho_{AR}$ is entangled and there exists a pure state $|\Phi_C\rangle\langle\Phi_C|$ such that $L_{BRC} \star |\Phi_C\rangle\langle\Phi_C| \in \mathcal{B}(\mathcal{H}_R \otimes \mathcal{H}_C)$ is proportional to an entangled state.*

*Proof.* Analogously to the proof of Prop. 1, it is straightforward to show that a separable initial state leads to a comb $\Upsilon_{ABC}$ that satisfies $\mathcal{E}(A:B|c) = 0$. Now, focusing on the second part of the proposition, we assume that $L_{BRC} \star |\Phi_C\rangle\langle\Phi_C|$ is separable for all pure states $|\Phi_C\rangle$. As any POVM element $E_C$ can – up to normalization – be written as a convex combination of pure states, this implies

$$\Upsilon_{AB|c} = \rho_{AR} \star L_{BRC} \star E_C^{\mathrm{T}} = \rho_{AR} \star \sum_i p_i \eta_R^{(i)} \otimes \xi_B^{(i)} = \sum_i p_i \, \mathrm{tr}[\rho_{AR} \eta_R^{(i)}] \otimes \xi_B^{(i)}, \qquad (22)$$

where $\eta_R^{(i)}, \xi_B^{(i)} \geq 0$ and $\{p_i\}$ are probabilities that add up to one. Evidently, the last expression in Eq. (22) is proportional to a separable state, which concludes the proof. $\qquad\square$

Note that $L_{BRC} \star |\Phi_C\rangle\langle\Phi_C| =: F_{BR}$ corresponds to a POVM element on $\mathcal{B}(\mathcal{H}_B \otimes \mathcal{H}_R)$. If it is entangled, then Charlie can 'entangle' $R$ and $B$ (and, in turn, possibly $A$ and $B$) by conditioning



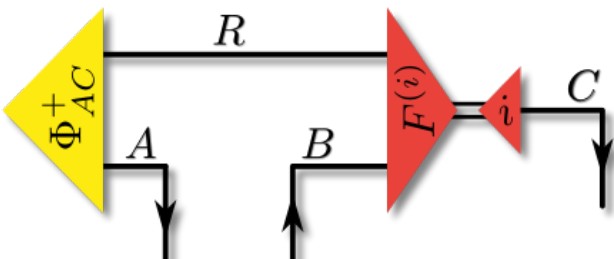

Figure 6: **Measurement in entangled basis.** The map $\mathcal{L}$ measures the state on $\mathcal{B}(\mathcal{H}_B \otimes \mathcal{H}_R)$ in the Bell basis and feeds forward 0 if the result corresponds to $\Phi^+_{BR}$ and 1 otherwise. If Charlie measures in the computational basis, conditioned on his outcome, Alice and Bob share a channel that can transmit quantum information (for outcome 0) or not (for outcome 1) from Bob to Alice.

on one of his outcomes, thus allowing Bob to send quantum information to Alice. In general, $F_{BR}$ is not proportional to the Choi matrix of a channel (i.e., a CPTP map), but of a trace non-increasing CP map. Consequently, the probability $p$ for Charlie to obtain an outcome corresponding to $\Phi_C$ depends on Bob's input state. On the other hand – as has been noted in [69] and, in a slightly different context, in [27, 74] – if $F_{BR} = q\widetilde{F}_{BR}$ is proportional to the Choi matrix of a CPTP map $\widetilde{F}_{BR}$, then this probability $p$ does *not* depend on Bob's input state $\tau_B$:

$$p = \mathrm{tr}[\rho_{AR} \star \tau_B \star L_{BRC} \star |\Phi_C\rangle\langle\Phi_C|] = q\,\mathrm{tr}[\widetilde{F}_{BR} \star (\rho_{AR} \otimes \tau_B)] = q\,, \tag{23}$$

where we have used the fact that $F_{BR}$ is CPTP, and $\{\rho_{AR} \otimes \tau_B\}$ are quantum states. To make this statement more concrete and to provide an explicit example, consider a variation of a teleportation scheme (without classical communication); let the system-environment map $\mathcal{L}$ be a measure and prepare channel of the form

$$\mathcal{L}[\rho_{BR}] = \mathrm{tr}(F^{(0)}_{BR}\rho_{BR})\,|0\rangle\langle 0|_C + \mathrm{tr}(F^{(1)}_{BR}\rho_{BR})\,|1\rangle\langle 1|_C\,, \tag{24}$$

where $\{F^{(i)}\}$ are POVM elements that sum to identity, and the initial system-environment state is the maximally entangled state (see Fig. 6). Choosing $F^{(0)} = |\Phi^+_{BR}\rangle\langle\Phi^+_{BR}|$, we see that if Charlie measures in the computational basis, then $\Upsilon_{AB|0} = \frac{1}{2}\Phi^+_{AB}$ and $\Upsilon_{AB|1} = \frac{1}{2}(\mathbb{1}_{AB} - \Phi^+_{AB})$. Both of them are proportional to CPTP maps. In particular, $\Upsilon_{AB|0}$ is proportional to the identity channel from Bob to Alice, while $\Upsilon_{AB|0}$ is proportional to an entanglement breaking channel. The probability of simulation is independent of Bob's input state. For example, we have

$$p(0) = \mathrm{tr}[(\Phi^+_{AR} \otimes \tau_B)(\Phi^+_{BR} \otimes \mathbb{1}_A)] = \frac{1}{4} \quad \forall \tau_B. \tag{25}$$

This simulation probability of 1/4 is the maximal achievable simulation probability for an identity channel from the future to the past [69]. Having the above resources at hand, Alice, Bob, and Charlie can thus simulate quantum channels from the future to the past (albeit *not* deterministically).

The necessary conditions for $\mathcal{E}(A:C|b) > 0, \mathcal{E}(B:C|a) > 0$, and $\mathcal{E}(A:B|c) > 0$ shed light on the underlying necessary resources as well as the intuitive interpretation of the respective cases. However, the requirements we found are not sufficient to ensure the existence of any of the three kinds of bipartite entanglement we discussed, and simple examples that satisfy the respective necessary conditions but do not display entanglement in the considered splitting can readily be constructed. Before advancing to the genuinely tripartite case, we now investigate bipartite entanglement from a more global perspective and link it to the entanglement breaking properties of certain maps.

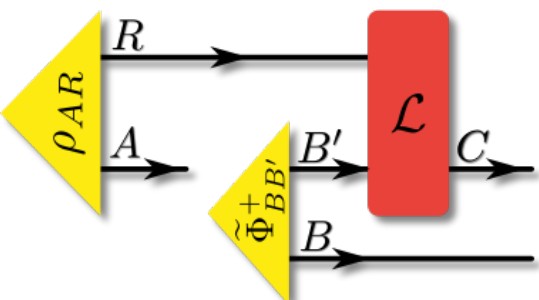

Figure 7: **CJI of a two-step process**. $\Upsilon_{ABC}$ is obtained by letting $\mathcal{L}$ act on one half of an unnormalized maximally entangled state and the degrees of freedom denoted by $R$. Note that we let $\mathcal{L}$ act on $\mathcal{B}(\mathcal{H}_{B'})$, so that $L_{BRC} \in \mathcal{B}(\mathcal{H}_B \otimes \mathcal{H}_R \otimes \mathcal{H}_C)$. As $\mathcal{H}_{B'} \cong \mathcal{H}_B$, this is merely a relabelling for notational purposes.

# 4 Sufficient condition for bipartite entanglement in AB:C and AC:B and channel steering

## 4.1 Sufficient condition for bipartite entanglement in processes for AB:C and AC:B

In the previous section we derived necessary conditions on a process not to be biseparable in some splitting. Here we will provide a sufficient condition on processes to be entangled in the splitting AB:C and AC:B. In particular, we will show that if a certain map is not entanglement breaking, one observes bipartite entanglement in $\Upsilon_{BC}$. As entanglement cannot be created by tracing out one party this implies that in this case the corresponding comb has to be entangled across the bipartite splittings AB:C and AC:B. More precisely, we consider the map

$$\Sigma[\omega] = \mathrm{tr}_R[\mathcal{L}(\rho_R \otimes \omega)], \tag{26}$$

where $\rho_R = \mathrm{tr}_A[\rho_{AR}]$ and $\mathcal{L}$ are given by the process. Then one can show the following Lemma:

**Lemma 4.** *Let $\Upsilon_{ABC}$ be a process defined by $\rho_{AR}$ and $\mathcal{L}$, $\Upsilon_{BC} = \mathrm{tr}_A(\Upsilon_{ABC})$ and $\Sigma$ be the CPTP map associated to this process via Eq. (26). Then $\Upsilon_{BC}$ is separable if and only if the map $\Sigma$ is entanglement breaking.*

*Proof.* Recall that (see also Fig. 7)

$$\Upsilon_{ABC} = \rho_{AR} \star L_{BRC} = \mathrm{tr}_R[\mathcal{L}(\rho_{AR} \otimes \tilde{\Phi}^+_{BB'})]. \tag{27}$$

Hence, we have that $\Upsilon_{BC} \propto \mathcal{I}_B \otimes \Sigma[|\Phi^+\rangle_{BB'}]$. This yields a separable state if and only if $\Sigma$ is an entanglement breaking map (see [55] and references therein) which proves the Lemma. $\quad\square$

Note that this implies that in case $\Sigma$ is not entanglement breaking $\Upsilon_{BC}$ is entangled and so is $\Upsilon_{ABC}$ across the bipartite splittings AB:C and AC:B. The converse is not necessarily true. By taking the partial trace entanglement may vanish and hence $\Upsilon_{ABC}$ may show bipartite entanglement even though $\Sigma$ is entanglement breaking. An example of a genuinely multipartite entangled process for which $\Upsilon_{BC}$ is separable is given in Eq. (50) with $n = 3$ (see Section 7).

In summary we have shown that if one disregards the initial state and inserts in the first time step a maximally entangled state and one still observes entanglement after the process took place then the quantum comb describing this dynamics has to be entangled in the bipartite splittings AB:C and AC:B.

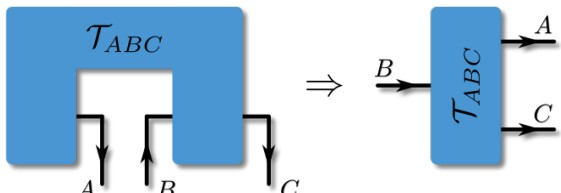

Figure 8: **Comb as a broadcasting channel.** Due to its properties, any comb $\mathcal{T}_{ABC}$ can be understood as a channel with one input and two output spaces $\mathcal{T}_{ABC} : \mathcal{B}(\mathcal{H}_B) \to \mathcal{B}(\mathcal{H}_C) \otimes \mathcal{B}(\mathcal{H}_A)$ with an additional trace condition (on its Choi state) that ensures causal ordering.

## 4.2 Channel steering and quantum combs

We finish our discussion of sufficient conditions for bipartite entanglement in combs with the case $\mathcal{E}(A : BC) > 0$. While the corresponding conditions on their own are not very illuminating, it is nonetheless insightful to consider this type of bipartite quantum correlations and connect it to the concept of *channel steering* [35].

To this end, consider a quantum channel $\mathcal{K} : \mathcal{B}(\mathcal{H}_B) \to \mathcal{B}(\mathcal{H}_A) \otimes \mathcal{B}(\mathcal{H}_C)$ with corresponding Choi matrix $K_{BAC}$ (such channels with one input and two output spaces are also called 'broadcasting channels' [75]). As $\mathcal{K}$ is assumed to be a channel, it satisfies $\mathrm{tr}_{AC}(K_{BAC}) = \mathbb{1}_B$. From Eq. (8), we see that any comb $\Upsilon_{ABC}$ also satisfies $\mathrm{tr}_{AC}(\Upsilon_{ABC}) = \mathbb{1}_B$ (with the additional constraint $\mathrm{tr}_C(\Upsilon_{ABC}) = \mathbb{1}_B \otimes \Upsilon_A$), and can thus be considered as a quantum broadcast channel (see Fig. 8). Consequently, while originally applied to general broadcast channels [35], all of the following considerations directly apply to combs.

Given such a channel $K_{BAC}$, Alice could measure the system $A$, using a POVM $\{(E_A^{a|x})^{\mathrm{T}}\}_a$, where $x$ denotes the choice of POVM, $a$ corresponds to the respective outcome, and the transpose is added to simplify the subsequent notation. Conditionally on each of Alice's outcomes, the mapping between Bob and Charlie would be given by (trace non-increasing) CP maps

$$K_{BC}^{a|x} := K_{BAC} \star E_A^{a|x} \tag{28}$$

that add up to a CPTP map $K_{BC} = \sum_a K_{BC}^{a|x}$. In this way, Alice could create a collection $\{K_{BC}^{x|a}\}_{a,x}$ of instruments that all add up to the same channel $K_{BC}$. Such a collection – in close analogy with the theory of steering of quantum *states* [13–15] – is called a 'channel assemblage' of the channel $K_{BC}$. A channel assemblage is said to be unsteerable, if there exists an instrument $\{K_{BC}^{\lambda}\}_{\lambda}$, probabilities $p(\lambda)$ and conditional probabilities $p(a|x, \lambda)$, such that for all $\{a, x\}$ one has[3]

$$K_{BC}^{a|x} = \sum_{\lambda} p(\lambda) p(a|x, \lambda) K_{BC}^{\lambda} . \tag{29}$$

Otherwise, the assemblage is steerable. If Alice can create a steerable assemblage, then the channel $K_{BAC}$ is said to be steerable. Evidently, channel steerability in the above sense is equivalent to steerability of the (normalized) state $K_{BAC}$ by Alice [35].

In our case, channel steerability implies that Alice, by performing measurements, can steer the mapping between Bob and Charlie. As entanglement is a prerequisite for steerability [13], here, entanglement of $\Upsilon_{ABC}$ in the splitting $A : BC$ is a prerequisite for Alice being able to steer Bob and Charlie. In line with our above considerations, we thus obtain the following Proposition:

---

[3]For simplicity, here we only consider the case of countably many indices $\lambda$. The generalization to more general sets is straightforward.

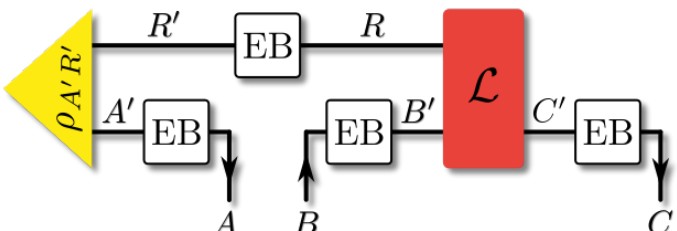

Figure 9: **Process with an entanglement breaking map on at least one of its spaces.** If the circuit of a process can be represented with an entanglement breaking (EB) channel on one of its wires, then the resulting comb $\Upsilon_{ABC}$ is separable in the corresponding cut. For example, an entanglement breaking channel on the environment $R$ implies that $\Upsilon_{ABC}$ is separable in the splitting $A : BC$. If there are two entanglement breaking channels (independent of what two wires they act on), then the resulting comb is fully separable. For better tracking of the involved spaces, the input and output spaces of the EB channels are labelled differently.

**Proposition 5.** *If the comb $\Upsilon_{ABC}$ is steerable by Alice acting on A, then the initial system environment state $\rho_{AR}$ is steerable by Alice acting on A.*

*Proof.* Using different instruments (labelled by $x$) with outcomes $a$, Alice can create a state assemblage $\{\rho_R^{a|x}\}_{a|x}$ of the state $\rho_R = \mathrm{tr}_A(\rho_{AR})$, i.e., $\rho_R^{a|x} = \rho_{AR} \star E_A^{a|x}$ and $\sum_a \rho_R^{a|x} = \rho_R$ for all choices of instruments $x$. Let us assume that the state $\rho_{AR}$ is unsteerable on Alice's side. This means that, for any state assemblage $\{\rho_R^{a|x}\}_{a|x}$, there exists a set $\{\rho_R^\lambda\}$ of subnormalized states, probabilities $p(\lambda)$ and conditional probabilities $p(a|x,\lambda)$, such that

$$\rho_R^{a|x} = \sum_\lambda p(\lambda)p(a|x,\lambda)\rho_R^\lambda. \tag{30}$$

Using this identity, we see that for any conditional mapping $\Upsilon_{BC}^{a|x}$ between Bob and Alice, we have

$$\Upsilon_{BC}^{a|x} = \rho_R^{x|a} \star L_{BRC} = \sum_\lambda p(\lambda)p(a|x,\lambda)\rho_R^\lambda \star L_{BRC} := \sum_\lambda p(\lambda)p(a|x,\lambda)L_{BC}^\lambda. \tag{31}$$

It is easy to check that $\{L_{BC}^\lambda\}$ constitutes an instrument, implying that any channel assemblage $\{\Upsilon_{BC}^{a|x}\}$ is unsteerable if $\rho_{AR}$ is unsteerable on Alice's side. This, in turn, means that if the broadcast channel $\Upsilon_{ABC}$ is steerable by Alice, then so is $\rho_{AR}$. $\qquad\square$

Having analysed necessary and sufficient conditions on the dynamical building blocks for entanglement in different splittings of combs, as well as the interpretation of these quantum correlations, we finish our discussion of the bipartite case, providing a connection between biseparable combs and entanglement breaking channels.

## 5 Separability and entanglement breaking channels

As we have seen in Sec. 4.1, separability of a comb $\Upsilon_{ABC}$ in the splittings $AB : C$ and $AC : B$ is directly related to the entanglement breaking property of a map (given by Eq. (26)) that arises naturally from the building blocks of the comb. Such a relation between the separability of a comb $\Upsilon_{ABC}$ in a given splitting and entanglement breaking (EB) maps within the circuit that yields said comb can be made more concrete, complementing the results of Sec. 4.1.

Specifically, here we investigate the question if the presence of an entanglement breaking map on one of the wires (see Fig. 9 for a graphical representation) implies separability of $\Upsilon_{ABC}$ in a given splitting, and vice versa. We give an affirmative answer for the splittings $C : AB$ and provide a partial resolution for the cases $A : BC$ and $B : AC$.

## 5.1 Entanglement breaking channels imply separable combs

First, it is straightforward to show that the presence of an EB channel on any of the wires leads to a comb $\Upsilon_{ABC}$ that is separable with respect to a particular splitting. Concretely, we will say that a comb $\Upsilon_{ABC}$ can be represented with an EB channel on one of its wires, if it admits a decomposition with an EB channel. For example, for the wire $R$, this would mean that $\Upsilon_{ABC}$ can be written as $\Upsilon_{ABC} = \rho_{AR'} \star N_{R'R}^{\text{EB}} \star L_{BRC}$ (see Fig. 9), where $N_{R'R}^{\text{EB}}$ is the Choi state of an entanglement breaking channel. With this, we have the following Proposition:

**Proposition 6.** *If a comb $\Upsilon_{ABC}$ can be represented with an entanglement breaking channel on one of its wires, then it is separable in at least one possible splitting. In particular, an entanglement breaking channel on A or R implies $\mathcal{E}(A : BC) = 0$, on B implies $\mathcal{E}(B : AC) = 0$, and on C implies $\mathcal{E}(C : AB) = 0$.*

The proof can be found in App. B. For the case of an EB channel on $R$, within the study of quantum memory, a similar Proposition has been shown in [32]. There, processes on two times that only display classical memory were defined as those that have an EB channel on $R$. Additionally, related results can be found in [34], where entanglement breaking superchannels and their representations are analysed. Here, we provide direct simple proofs that will also allow us to show the converse for one of the cases and to pinpoint the difficulties with proving the converse of the other two.

From the above Proposition, it is evident, that a circuit with at least two entanglement breaking channels in its representation (as long as they are not on $A$ and $R$) is separable with respect to two distinct splittings. While this fact in itself does not yet imply that it is fully separable, we have the following Lemma:

**Lemma 7.** *If a circuit can be represented with an EB channel on any two of the wires $\{A/R, B, C\}$ it is fully separable.*

The proof can be found in App. C. While an EB channel on one of the wires always leads to a comb that is separable in the corresponding splitting, it is *a priori* unclear, if every separable comb $\Upsilon_{ABC}$ has a representation that contains an EB channel on the correct wire.

## 5.2 $\mathcal{E}(C : AB) = 0$ implies EB channel on C

First, consider the case $\mathcal{E}(C : AB) = 0$. We have the following Proposition:

**Proposition 8.** *If a proper comb $\Upsilon_{ABC}$ satisfies $\mathcal{E}(C : AB) = 0$, then there exists an initial state $\rho_{AR}$, a CPTP map $L_{BRC'}$ and an EB channel $N_{C'C}^{EB}$, such that $\Upsilon_{ABC} = \rho_{AR} \star L_{BRC'} \star N_{C'C}^{EB}$*

*Proof.* If $\mathcal{E}(C : AB) = 0$, then $\Upsilon_{ABC}$ is of the form

$$\Upsilon_{ABC} = d_B \sum_{\alpha} p(\alpha) \xi_{AB}^{(\alpha)} \otimes \eta_C^{(\alpha)}, \tag{32}$$

where $\{p(\alpha)\}$ are probabilities that sum up to one, $\{\xi_{AB}^{(\alpha)}, \eta_C^{(\alpha)}\}$ are states on the respective spaces, and $d_B = \dim(\mathcal{H}_B)$. Here, and in the following proofs, for clarity of the exposition, we

make the normalization of the comb evident by factoring out the factor $d_B$. If $\Upsilon_{ABC}$ is a proper comb, then so is

$$\widetilde{\Upsilon}_{ABC'} = d_B \sum_\alpha p(\alpha) \xi_{AB}^{(\alpha)} \otimes |\alpha\rangle\langle\alpha|_{C'} \, , \tag{33}$$

where $\langle\alpha|\alpha'\rangle_{C'} = \delta_{\alpha\alpha'}$. Now, choosing $N_{C'C}^{\mathrm{EB}} = \sum_\alpha |\alpha\rangle\langle\alpha|_{C'} \otimes \eta_C^{(\alpha)}$, we can concatenate $\widetilde{\Upsilon}_{ABC'}$ and $N_{C'C}^{\mathrm{EB}}$ to obtain

$$
\begin{aligned}
\widetilde{\Upsilon}_{ABC'} \star N_{C'C}^{\mathrm{EB}} &= \sum_\alpha \langle\alpha_{C'}|\widetilde{\Upsilon}_{ABC'}|\alpha_{C'}\rangle \otimes \rho_C^{(\alpha)} \\
&= \Upsilon_{ABC} \, ,
\end{aligned}
\tag{34}
$$

which implies that $\Upsilon_{ABC}$ can be understood as coming from a process given by $\widetilde{\Upsilon}_{ABC'}$, with an additional entanglement breaking channel on the final output leg $C'$. As $\Upsilon_{ABC}$ is a proper comb, there is an underlying circuit with initial state $\rho_{AR}$ and CPTP map $L_{BRC'}$ that leads to it. This concludes the proof. □

While the above Proposition provides a constructive way to obtain a circuit with an entanglement breaking map on $C$ for any comb $\Upsilon_{ABC}$ that is separable in the splitting $C : AB$, it comes with a caveat. In principle, depending on how many terms make up $d_B \sum_\alpha p(\alpha) \rho_{AB}^{(\alpha)} \otimes \eta_C^{(\alpha)}$, the dimension of $C'$ can be much bigger than the dimension of $C$. It remains an open question, if there is also always a realization with an EB channel that satisfies $\dim(\mathcal{H}_{C'}) = \dim(\mathcal{H}_C)$.

Before advancing, let us emphasize the operational meaning of the absence of entanglement in the $C : AB$ splitting in terms of the action of the corresponding comb on a bipartite state $\rho_{BB'}$ that is fed into the process. The resulting tripartite state on $AB'C$ is given by

$$\Upsilon_{ABC} \star \rho_{BB'} = d_B \sum_\alpha p(\alpha)(\xi_{AB}^{(\alpha)} \star \rho_{BB'}) \otimes \eta_C^{(\alpha)} =: \sum_\alpha p(\alpha)(\kappa_{AB'}^{(\alpha)} \otimes \eta_C^{(\alpha)}), \tag{35}$$

which is separable in the splitting $C : AB'$, but potentially entangled in the $A : BC'$ splitting. Consequently, any comb that satisfies $\mathcal{E}(C : AB) = 0$ breaks the original entanglement of *any* state $\rho_{BB'}$ that is fed into the process (otherwise, entanglement between $B'$ and $C$ would have to be present in the above state), but potentially entangles it with another system (here, the system $A$). In this sense, entanglement of an input state can merely be swapped, but not be distributed between all three parties. We will see below that similar operational statements also apply to combs that are separable in different splittings.

## 5.3 $\mathcal{E}(A : BC) = 0$ and EB channels on A

The case of combs that are separable in the splitting $A : BC$ has been discussed in [32] as a potential superset of the set of processes that only display classical memory (i.e., processes where only classical information is fed forward on the environment line $R$), and the question was left open, if it constitutes a proper superset. Phrased in our nomenclature, this question amounts to the question if every comb that is separable with respect to the splitting $A : BC$ can be represented as a comb with an EB channel on $A$ (or, equivalently, $R$). Here, we provide a partial answer to this question. To do so, we need the following Lemma:

**Lemma 9.** *Any proper comb of the form* $\Upsilon_{ABC} = \sum_\alpha \rho_A^{(\alpha)} \otimes G_{BC}^{(\alpha)}$, *where* $\{\rho_A^{(\alpha)}\}$ *is a set of quantum states and* $G_{BC}^{(\alpha)} \geq 0$ *for all* $\alpha$ *can be represented with an EB channel on A if all* $G_{BC}^{(\alpha)}$ *are proportional to CPTP maps.*

*Proof.* By assumption, we have $\mathrm{tr}_C(G_{BC}^{(\alpha)}) = \mu_\alpha \mathbb{1}_B$, where $\mu_\alpha > 0$ (as $G_{BC}^{(\alpha)} \geq 0$). From $\mathrm{tr}_{AC}(\Upsilon_{ABC}) = \mathbb{1}_B$, it follows that $0 < \mu_\alpha \leq 1$ and $\sum_\alpha \mu_\alpha = 1$, implying that $\{\mu_\alpha\}_\alpha$ is a probability distribution. Now, choose an initial system-environment state

$$\rho_{AR} = \sum_\alpha \mu_\alpha \rho_A^{(\alpha)} \otimes |\alpha\rangle\langle\alpha| , \tag{36}$$

with $\langle\alpha|\alpha'\rangle = \delta_{\alpha\alpha'}$, and a map $L_{BRC} = \sum_\alpha |\alpha\rangle\langle\alpha|_R \otimes \widetilde{G}_{BC}^{(\alpha)}$, where $\widetilde{G}_{BC}^{(\alpha)}$ is the CPTP map $G_{BC}^{(\alpha)}/\mu_\alpha$. Intuitively, $L_{BRC}$ is the Choi matrix of a map that measures the environment $R$ in the computational basis and, controlled on the outcome performs the CPTP map $\widetilde{G}_{BC}^{(\alpha)}$ on $B$. As the channel $L_{BRC}$ given above is positive and satisfies $\mathrm{tr}_C(L_{BRC}) = \mathbb{1}_{BR}$ it is a CPTP map. With these choices, by construction we have $\Upsilon_{ABC} = \rho_{AR} \star L_{BRC}$. Now, defining $\widetilde{\rho}_{A'R} = \sum_\alpha |\alpha\rangle\langle\alpha|_{A'} \otimes |\alpha\rangle\langle\alpha|_R$, and an EB channel on $A'$ of the form $N_{AA'}^{\mathrm{EB}} = \sum_\alpha |\alpha\rangle\langle\alpha|_{A'} \otimes \rho_A^{(\alpha)}$, it is easy to see that $\Upsilon_{ABC} = \widetilde{\rho}_{A'R} \star N_{AA'}^{\mathrm{EB}} \star L_{BRC}$. $\square$

While in the above proof, we consider the case of representing the comb $\Upsilon_{ABC}$ by means of an EB channel on $A$, in closer correspondence with the considerations of [32], we could also have considered a representation by means of an EB channel on $R$. Concretely, choosing an EB channel $N_{RR'}^{\mathrm{EB}} = \sum_\alpha |\alpha\rangle\langle\alpha|_{R'} \otimes |\alpha\rangle\langle\alpha|_R$ on the environment, one sees that, with the appropriate relabeling, $\rho_{AR} = N_{RR'}^{\mathrm{EB}} \star \rho_{AR'}$, where $\rho_{AR} \cong \rho_{AR'}$, implying that inserting the EB channel $N_{RR'}^{\mathrm{EB}}$ on $R$ does not change the resulting comb $\Upsilon_{ABC}$.

Using Lem. 9, we can show that a proper subset of combs that are separable in the $A:BC$ cut admit a representation that includes an EB channel on $A$.

**Proposition 10.** *A comb of the form* $\Upsilon_{ABC} = \sum_\alpha \rho_A^{(\alpha)} \otimes G_{BC}^{(\alpha)}$, *where* $\{\rho_A^{(\alpha)}\}$ *is a set of quantum states and* $G_{BC}^{(\alpha)} \geq 0$ *for all* $\alpha$, *can be represented by a circuit containing an EB channel on* $A$ *if all states in* $\{\rho_A^{(\alpha)}\}_\alpha$ *are linearly independent in* $\mathcal{B}(\mathcal{H}_A)$.

*Proof.* Let $\{E_A^{(j)\mathrm{T}}\}_j$ be an informationally complete POVM on $A$. If Alice performs a measurement on her system, with an outcome corresponding to the POVM element $E_A^{(j)\mathrm{T}}$, then the resulting mapping $F_{BC}^{(j)}$ between Bob and Charlie is given by

$$F_{BC}^{(j)} = E_A^{(j)} \star \Upsilon_{ABC} . \tag{37}$$

Up to a normalization constant, given by the probability $q_j = 1/d_B \cdot \mathrm{tr}(E_A^{(j)} \star \Upsilon_{ABC})$, $F_{BC}^{(j)}$ is a CPTP map, and we set $F_{BC}^{(j)} = q_j \widetilde{F}_{BC}^{(j)}$. This property can now be used to show that every $G_{BC}^{(\alpha)}$ is proportional to a CPTP map. To this end, we remark that for every set of linearly independent matrices $\{\rho_A^{(\alpha)}\}$, there exists a set of *dual* matrices $\{\Delta_A^{(\beta)}\}$, such that $\mathrm{tr}(\rho_A^{(\alpha)} \Delta_A^{(\beta)\dagger}) = \delta_{\alpha\beta}$ [38,76]. If all matrices $\{\rho_A^{(\alpha)}\}$ are Hermitian, then so are all the duals. However, they are in general not positive – even if they are the duals of positive matrices – making their physical interpretation difficult. As they are Hermitian, each dual can be written as a real linear combination $\Delta_B^{(\beta)} = \sum_j c_j^{(\beta)} E_A^{(j)\mathrm{T}}$ of the informationally complete POVM elements $\{E_A^{(j)\mathrm{T}}\}_j$. With this, we have

$$G_{BC}^{(\alpha)} = \mathrm{tr}(\Upsilon_{ABC} \Delta_A^{(\alpha)\dagger}) = \sum_j c_j^{(\alpha)} E_A^{(j)} \star \Upsilon_{ABC} = \sum_j c_j^{(\alpha)} q_j \widetilde{F}_{BC}^{(j)} . \tag{38}$$

Consequently, for all $\alpha$, $\mathrm{tr}_C(G_{BC}^{(\alpha)}) = \sum_j c_j^{(\alpha)} q_j \mathbb{1}_B$ holds, which implies that all matrices in $\{G_{BC}^{(\alpha)}\}$ are proportional to CPTP maps. By Lem. 9 the comb $\Upsilon_{ABC}$ can then be represented with an EB channel on $A$. $\square$

As for the previous Lemma, the above proof also holds for an EB channel on $R$. Importantly, though, not every $\Upsilon_{ABC}$ that is separable in the $A:BC$ splitting can necessarily be represented by means of linearly independent states $\{\rho_A^{(\alpha)}\}_\alpha$. If this requirement is not fulfilled, then the matrices $G_{BC}^{(\alpha)}$ can not be 'addressed' anymore by means of duals, and the above proof fails to work. In particular, as causality constraints only require that $\sum_\alpha G_{BC}^{(\alpha)}$ is CPTP, there might exist proper combs $\Upsilon_{ABC}$ that can be decomposed as $\Upsilon_{ABC} = \sum_\alpha \rho_A^{(\alpha)} \otimes G_{BC}^{(\alpha)}$, but cannot be represented as a separable matrix with all terms on $BC$ proportional to CPTP maps.

As at the end of the previous section, we can, again, give an operational meaning to separability of a comb in the $A:BC$ splitting in terms of its action on an input state $\rho_{BB'}$. By direct insertion, we see that for any input state $\rho_{BB'}$ the resulting tripartite state $\Upsilon_{ABC} \star \rho_{BB'}$ is separable in the $A:B'C$ splitting and can thus at most be entangled between $B'$ and $C$, but not between $B'$ and $A$. This, in turn implies, that a process described by a comb that satisfies $\mathcal{E}(A:BC) = 0$ allows one to pertain entanglement, but cannot entangle input states with other degrees of freedom.

## 5.4 $\mathcal{E}(B:AC) = 0$ and EB channels on B

As for the case of combs that are separable in the splitting $A:BC$, the causality constraints that $\Upsilon_{ABC}$ has to satisfy only allow for a partial result concerning the connection of EB channels on $B$ and the separability of $\Upsilon_{ABC}$ in the splitting $B:AC$. We have the following Proposition:

**Proposition 11.** *A comb of the form $\Upsilon_{ABC} = d_B \sum_\alpha p(\alpha) E_B^{(\alpha)} \otimes G_{AC}^{(\alpha)}$, with $E_B^{(\alpha)}, G_{AC}^{(\alpha)} \geq 0$ and $\mathrm{tr}(E_B^{(\alpha)}) = \mathrm{tr}(G_{AC}^{(\alpha)}) = 1$ can be represented by a circuit containing an EB channel on $B$ if all matrices in $\{E_B^{(\alpha)}\}_\alpha$ are linearly independent.*

*Proof.* First, using the causality constraints on $\Upsilon_{ABC}$, we see that the reduced state in Alice's lab is independent of the state that Bob feeds into the process, i.e., for any quantum states $\{\tau_B, \tau_B'\}$,

$$\Upsilon_{ABC} \star \tau_B \star \mathbb{1}_C = \Upsilon_{ABC} \star \tau_B' \star \mathbb{1}_C =: \rho_A \tag{39}$$

holds, where $\mathbb{1}_C$ is the Choi matrix of the operation that discards the system, the only trace preserving operation Charlie can perform [68]. Additionally, the causality constraint implies

$$d_B \sum_\alpha p(\alpha) E_B^{(\alpha)} = \mathbb{1}_B. \tag{40}$$

Now, analogous to the proof of Prop. 10, let $\{\Delta_B^{(\beta)}\}$ be the dual set of $\{E_B^{(\alpha)}\}$. Choosing a set of $d_B^2$ linearly independent states $\{\tau_B^{(j)}\}$, each dual can be written as $\Delta_B^{(\alpha)\dagger} = \sum_j c_j^{(\alpha)} \tau_B^{(j)}$. From Eq. (40), it is easy to see that

$$\mathrm{tr}(\Delta_B^{(\alpha)\dagger}) = \sum_j c_j^{(\alpha)} = d_B p(\alpha). \tag{41}$$

With this, we have

$$G_{AC}^{(\alpha)} = \frac{1}{d_B p(\alpha)} \Upsilon_{ABC} \star \Delta_B^{(\alpha)*} = \frac{1}{d_B p(\alpha)} \sum_j c_j^{(\alpha)} \Upsilon_{ABC} \star \tau_B^{(j)*} =: \frac{1}{d_B p(\alpha)} \sum_j c_j^{(\alpha)} \rho_{AC}^{(j)}, \tag{42}$$

where the complex conjugate is required to comply with the definition of the link product. From Eq. (39) it follows that all $\rho_{AC}^{(j)}$ have the same reduced state, i.e., $\mathrm{tr}_C(\rho_{AC}^{(j)}) = \rho_A$. Together

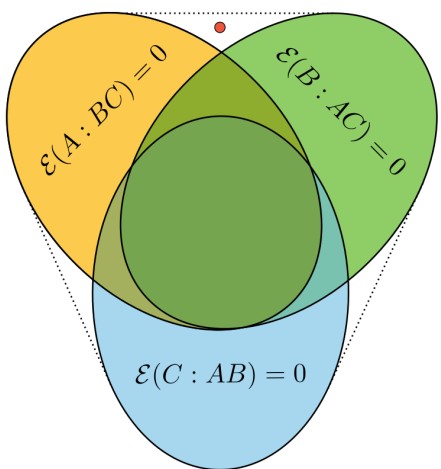

Figure 10: **Different sets of separable combs.** All states outside the convex hull of separable combs are GME. As we show in the main text, there are processes that are not bi-separable, but also not GME (the red dot in the figure would correspond to such a process.

with the fact that $\text{tr}(G_{AC}^{(\alpha)}) = 1$ and Eq. (41), this implies that $\text{tr}_C(G_{AC}^{(\alpha)}) = \rho_A$ for all $\alpha$. Then, it is easy to see that the matrix

$$\widetilde{\Upsilon}_{AB'C} = \sum_\alpha |\alpha\rangle\langle\alpha|_{B'} \otimes G_{AC}^{(\alpha)} \tag{43}$$

is also a proper comb, as it is positive and satisfies the causality constraints of Eq. (8). Now, defining the EB channel $N_{BB'}^{\text{EB}} = d_B \sum_\alpha p(\alpha) E_B^{(\alpha)} \otimes |\alpha\rangle\langle\alpha|_{B'}$, one sees that

$$\Upsilon_{ABC} = \widetilde{\Upsilon}_{AB'C} \star N_{BB'}^{\text{EB}}. \tag{44}$$

This concludes the proof. □

Again, as for the other two possible separable combs, there is a direct operational meaning to separability in the $B : AC$ splitting. Specifically, it implies that the resulting tripartite state $\Upsilon_{ABC} \star \rho_{BB'}$ is separable in the $B' : AC$ splitting for any input state $\rho_{BB'}$. In turn, this means that the corresponding process breaks any entanglement of its input states (as $B'$ is not entangled with the other degrees of freedom of the output state), but can be a source of entanglement between different degrees of freedom (here, $A$ and $C$).

Having discussed the bipartite case in detail, we now turn to the case of genuine multipartite entanglement, and its structure for the case of quantum processes.

## 6 Example of a biseparable process which shows non-zero bipartite entanglement in all splittings

Any genuinely multipartite entangled process has to be in the intersection between $\mathcal{E}(A : BC) > 0$, $\mathcal{E}(C : AB) > 0$ and $\mathcal{E}(B : AC) > 0$ as by definition it cannot be separable with respect to some bipartition. However, the conditions $\mathcal{E}(A : BC) > 0$, $\mathcal{E}(C : AB) > 0$ and $\mathcal{E}(B : AC) > 0$ do not imply genuine multipartite entanglement. This is well known (and straightforward to see) for states and holds also true for processes as the following example shows. The process

$$K_{ABC} = 2p |0\rangle\langle0|_A \otimes |\Phi^+\rangle\langle\Phi^+|_{BC} + (1-p)\mathbb{1}_B \otimes |\Phi^+\rangle\langle\Phi^+|_{AC} \tag{45}$$

is not genuinely multipartite entangled. However, for $1 > p > 1/3$ it has the following property: $\mathcal{E}(A:C) > 0$ and $\mathcal{E}(B:C) > 0$, (as the reduced states are not PPT). This implies that also $\mathcal{E}(A:BC) > 0$ and $\mathcal{E}(B:AC) > 0$. Moreover, the process is also not PPT with respect to the splitting $C:AB$ and hence $\mathcal{E}(C:AB) > 0$. This shows that there exist biseparable processes in the intersection of $\mathcal{E}(A:BC) > 0$, $\mathcal{E}(C:AB) > 0$ and $\mathcal{E}(B:AC) > 0$. Analogously, it also implies that $\mathcal{E}(A:C) > 0$ and $\mathcal{E}(B:C) > 0$ do not guarantee GME for processes either (note that $\mathcal{E}(A:B) = 0$ for any process due to the causality constraints). While we have that for this process $\mathcal{E}(A:C|b) > 0$ and $\mathcal{E}(B:C|a) > 0$ (for e.g. trivial measurements on the respective parties), it is straightforward to see that there exists no measurement such that $\mathcal{E}(A:B|c) > 0$.

One may wonder whether the intersection of $\mathcal{E}(A:B|c) > 0$, $\mathcal{E}(A:C|b) > 0$ and $\mathcal{E}(B:C|a) > 0$ contains also biseparable processes and the answer is yes. An example for such a biseparable process is given by

$$
\begin{aligned}
K_{ABC} =&\, 2p\,|0\rangle\langle 0|_A \otimes |\Phi^+\rangle\langle\Phi^+|_{BC} + p\,\mathbb{1}_B \otimes |\Phi^+\rangle\langle\Phi^+|_{AC} + \frac{1-2p}{2}(|\Phi^+\rangle\langle\Phi^+|_{AB} \otimes |0\rangle_C\langle 0| \\
&+ |\Phi^-\rangle\langle\Phi^-|_{AB} \otimes |1\rangle\langle 1|_C + |010\rangle\langle 010| + |101\rangle\langle 101|),
\end{aligned}
\tag{46}
$$

with $1/12(\sqrt{33} - 3) < p < 1/2$. That this process is indeed in the intersection can be easily proven by considering the post-measurement state obtained by measuring $|0\rangle\langle 0|_X$ for $X \in \{A, B, C\}$ respectively and showing that its partial transpose has a negative eigenvalue, i.e., it is entangled.

# 7 Genuinely multipartite entangled processes

So far we investigated conditions on processes to show (conditional) bipartite entanglement. In this section we will consider genuinely multipartite entangled processes, i.e., the set of processes that lie outside the convex hull of biseparable ones. This discussion is similar in spirit to the one conducted in [30, 31], where processes that cannot be understood as a mixture of direct and common cause processes were analysed. Our emphasis, however, lies on the genuine multipartite entanglement properties.

In particular, we will focus on the three-qubit case and we will provide examples of processes within each of the SLOCC classes, more precisely an example inside the W-class and one for a process in GHZ\W. This implies that (at least in the three qubit case) all types of mixed-state entanglement can occur for processes. Let us finally note for completeness that also biseparable processes that are not biseparable with respect to some specific splitting but some mixtures of such states can occur (see Section 6).

The first example of a genuinely multipartite entangled process is given by

$$
\Upsilon_{ABC} = |s_1\rangle\langle s_1| + |s_2\rangle\langle s_2| ,
\tag{47}
$$

with

$$
|s_1\rangle = \frac{|001\rangle}{\sqrt{2}} + \frac{|010\rangle + |100\rangle}{2} \quad \text{and} \quad |s_2\rangle = \frac{|110\rangle}{\sqrt{2}} + \frac{i(|101\rangle - |011\rangle)}{2}.
\tag{48}
$$

The fact that this process is indeed GME can be straightforwardly checked by means of the SDP in Eq. (12). Additionally, it constitutes a valid process due to

$$
\mathrm{tr}_C(\Upsilon_{ABC}) = \frac{\mathbb{1}}{2} \otimes \mathbb{1}.
\tag{49}
$$

As the comb of Eq. (47) satisfies the causality constraints of (8), there exists a quantum circuit with a pure initial state and a unitary system-environment dynamics that leads to it. We

provide this circuit in App. D and show explicitly that its building blocks satisfy the properties laid out in the previous sections.

The process given in Eq. (47) has the following properties: a) It has rank 2, which is the smallest possible rank for a genuinely multipartite entangled process. The latter can be easily seen by noting that there exists no genuinely multipartite entangled pure three-qubit state for which the causality constraint can be fulfilled (as already discussed in Sec. 2.2). b) It is in the W-class as it is a convex combination of two pure states within the W-class. c) It can easily be shown that this process is in the intersection of $\mathcal{E}(A : B|c) > 0$, $\mathcal{E}(C : A|b) > 0$ and $\mathcal{E}(B : C|a) > 0$. Note that this is not necessarily true for any GME process. In Sec. 8 and App. G we provide an example for a process with vanishing conditional entanglement for all splittings.

The second example is a construction of genuinely multipartite entangled processes for $n$ qubits and for which there exists in the three-qubit case no decomposition into pure fully separable, biseparable and W-class states, i.e., this process is not in the W-class. As we will show next for arbitrary $n$ the following processes are genuinely multipartite entangled:

$$
\begin{aligned}
\Upsilon^{(n)} = {} & \frac{1}{2^{\frac{n-3}{2}}} |GHZ_n\rangle\langle GHZ_n| \\
& + \frac{1}{2^{\frac{n-1}{2}}} (\mathbb{1}_{n-1} - |0\ldots0\rangle\langle0\ldots0| - |1\ldots1\rangle\langle1\ldots1|) \otimes |0\rangle\langle0|,
\end{aligned}
\tag{50}
$$

where $|GHZ_n\rangle = 1/\sqrt{2}(|0\ldots0\rangle + |1\ldots1\rangle)$ and $\mathbb{1}_k$ is the identity acting on $k$ qubits. It can be straightforwardly checked that they are valid processes as performing the partial trace over the last qubit results in

$$
\mathrm{tr}_n(\Upsilon^{(n)}) = \frac{\mathbb{1}_{n-1}}{2^{\frac{n-1}{2}}}
\tag{51}
$$

and hence all causality constraints are satisfied. Note that also all other marginals are separable. This implies that even though the process is genuinely multipartite entangled, all entanglement vanishes as soon as one of the parties is discarded. In order to show that these processes are genuinely multipartite entangled we use a necessary condition for biseparability first introduced in [61] which states that any biseparable $n$-qubit state, $\rho^{(n)}$ fulfills (see Sec. 2.2 for further details)

$$
|\rho^{(n)}_{(0\ldots0,1\ldots1)}| \le \frac{1}{2} \sum_{|I|\in\{1,\ldots,n-1\}} \sqrt{\rho^{(n)}_{(I,I)}\rho^{(n)}_{(\bar{I},\bar{I})}}.
\tag{52}
$$

As can easily be seen it holds for $\Upsilon^{(n)}$ that $\Upsilon^{(n)}_{(I,I)}\Upsilon^{(n)}_{(\bar{I},\bar{I})} = 0$ for $|I| \in \{1,\ldots,n-1\}$. However, we have that $|\Upsilon^{(n)}_{(0\ldots0,1\ldots1)}| = \frac{1}{2^{\frac{n-1}{2}}}$. Therefore, the necessary condition for biseparability in Eq. (52) is violated and the state is genuinely multipartite entangled.

Moreover, it should be noted that for $n = 3$ this process has non-zero tangle $\tau$ (see App. E) and is therefore not in the W-class. Hence, all different types of mixed three-qubit entanglement can occur for processes. Note that for the process it holds that $\mathcal{E}(A : C) = 0$ and $\mathcal{E}(B : C) = 0$ in spite of being genuinely multipartite entangled. Note further that it can be easily verified the state is in the intersection of $\mathcal{E}(A : B|c) > 0$, $\mathcal{E}(C : A|b) > 0$ and $\mathcal{E}(B : A|c) > 0$.

In App. F we provide a further simple example of a genuinely multipartite entangled comb on four parties. Specifically, the corresponding circuit only requires the two-fold application of $\sqrt{\mathcal{S}}[\rho] = \sqrt{S}\rho\sqrt{S}^\dagger$, where $S$ is the Swap gate (see Fig. 11). In the appendix we show that the corresponding comb is indeed genuinely multipartite entangled.

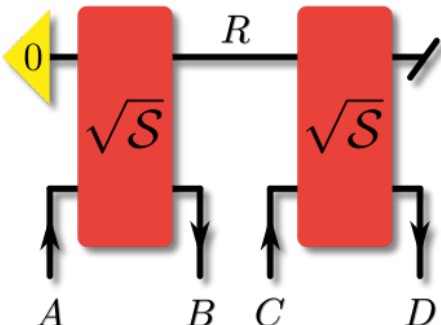

Figure 11: **Circuit that leads to a four-partite GME comb.** Note that using a Swap gate instead of $\sqrt{S}$ would *not* yield a GME comb (see App. F.)

Finally, returning to the operational meaning of entanglement in combs, as alluded to in Sec. 2.3, it is clear that a GME comb is a resource that allows one to transform bipartite entanglement, present in an input state $\rho_{BB'}$, to genuine multipartite entanglement. This, naturally, extends to the arbitrary times, in the sense that, starting from $k-1$ bipartite entangled states, a GME comb on $k$ times enables the creation of a genuinely multipartite quantum state.

## 8 GME Processes with no creatable entanglement

We have seen in Sec. 6 that there exist processes that are entangled in any bipartition, yet not genuinely multipartite entangled. While *all* GME processes are entangled in any bipartite splitting, the same does not hold true for their reduced processes; in particular, we have $\mathrm{tr}_C(\Upsilon_{ABC}) = \mathbb{1}_B \otimes \Upsilon_A$. However, as discussed in Sec. 3, there can be conditional entanglement in any of the reduced processes. This begs the question of whether there are GME processes that do not display conditional entanglement in *any* of their reduced versions, *i.e.*, GME processes where no party can induce entanglement between the remaining two parties by means of a measurement.

This question has been answered for states in [33], where a GME state with separable conditional marginals was provided, and, in addition, it was shown that the GME of the state can be detected from the separable marginals. As such a state is GME but does not even conditionally display any entanglement in its marginals, it has no creatable entanglement in its subsystem. Here, we investigate whether this phenomenon also exists for processes, i.e., whether there are proper combs $\Upsilon_{ABC}$ that are GME but have no creatable entanglement (this situation is depicted in Fig. 12).

To find such a process – defined on three qubit Hilbert spaces $\mathcal{H}_A$, $\mathcal{H}_B$, and $\mathcal{H}_C$ – we follow the procedure detailed in [33]; there, GME states with vanishing conditional entanglement have been found by means of a see-saw SDP. Starting from some initial state $\rho_0$, first, an optimal witness for GME is found by means of the SDP in Eq. (12). Then, employing a second SDP, an optimal GME state is found for this witness, with the additional requirement, that the conditional marginals of said state are separable for a 'sufficient' ($\sim 1000$ in [33]) number of projective measurements. More precisely, the conditional marginals of said state (with respect to the projective measurements) are required to be *strictly* positive under partial transpose. Iterating this procedure yields a good guess $\rho$ for a GME state with vanishing conditional entanglement, and it can then be shown numerically that the found state actually possesses the desired properties (see [33] for more details on the employed algorithm). Here, we can use the same algorithm, with the additional constraint that the 'states' we consider satisfy the causality constraint of Eq. (8). With this, we find a candidate process, for which we can verify

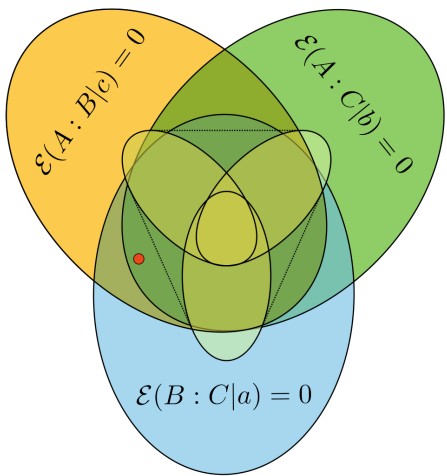

Figure 12: **GME combs without creatable entanglement.** In the main text, we provide a comb that lies outside the convex hull of separable combs, i.e., it is GME, but does not display any conditional entanglement. Exemplarily, this processes is depicted by a red dot.

numerically that it has vanishing conditional entanglement all measurements. We provide this process in App. G.

Given that an action of any of the parties destroys it, we conclude that the GME such processes possess is somewhat inaccessible; if Bob feeds forward any state, Alice and Charlie do not share an entangled state, while any measurement on Alice's (Charlie's) system will only allow for an entanglement breaking channel between Bob and Charlie (Alice).

# 9   Conclusion and outlook

In this work – using the quantum comb framework – we studied the entanglement properties of temporal processes. Despite the vast body of work on the structure and stratification of bi- and multipartite entanglement, due to the constraints imposed by causality, *a priori*, it had not been clear what types of entanglement and well known phenomena from entanglement theory can also persist in the temporal case. Here, we provided an overview of the types of entanglement that can occur in quantum combs, and furbished these different types of temporal quantum correlations with clear-cut operational interpretations; both in terms of the implications for the properties of the corresponding temporal process, and in terms of the building blocks of the underlying quantum circuits.

First, we derived both necessary and sufficient conditions for bipartite (conditional or unconditional) entanglement and discussed its relation to entanglement-breaking channels and channel steering (for the case $\mathcal{E}(BC : A) > 0$). Based on these results, a more refined notion of entanglement for temporal processes suggests itself. While here we considered the standard definition of entanglement, i.e., based on convex combinations of product states, for processes an alternative definition in terms of product *processes* seems equally reasonable. Then, a process would be entangled if it could not be written as a convex combination of processes that factorize in at least one splitting. Investigating the operational and structural implication of such a more fine-grained definition of entanglement for processes is subject to future research.

Next, we demonstrated that there exist genuinely multipartite entangled processes for processes with an arbitrary number of steps, and all types of three-qubit mixed state entanglement

can occur in the case where only two times are considered. Our investigation showed that there exist genuinely multipartite entangled processes with separable marginals. Finally, we provided an example of a GME process that does not display creatable entanglement, mirroring analogous results for the case of spatial entanglement.

Along the way, we related the separability of quantum combs in different splittings to the presence of entanglement breaking channels in their circuit representation. While the presence of an entanglement breaking channel in the underlying dynamics leads to a separable comb, the converse is not necessarily true for all possible splittings. We provided partial results on when the converse actually holds. Besides this interpretation in dynamical terms, we also gave direct operational interpretations of separable combs. Depending on the splitting, separable combs break and/or swap entanglement of bipartite input states, thus extending the corresponding results for the case of quantum channels to the multi-time case.

Our research sheds light on the interplay between the underlying physical process and the entanglement properties of the corresponding comb. There are two follow-up questions that suggest themselves: Can entanglement in time be understood on a deeper structural level, and how can genuine quantum correlations in time be exploited for the enhancement of quantum information processing tasks?

The former question concerns the peculiar structure of how entanglement in time is 'created'. The only system that allows for the interaction between non-adjacent parties is the environment, which, in the end, is discarded. The environment 'collides' with all the parties, and in doing so, transmits correlations between them. Such collision models have, for example, been used to investigate the creation of multipartite (spatial) entanglement [77]. The entanglement structure of such processes inevitably will affect their quantum complexity. A detailed investigation of the implications for the strength and distribution that entanglement in combs can display is subject to future work.

Considering the latter question, we have seen that many of the observed entanglement phenomena also exist in the spatial setting. It is important though, to emphasize that combs describe fundamentally different experimental scenarios than quantum states; quantum states allow for the computation of correlations for the case where spatially separated, non-communicating parties perform independent measurements. Quantum combs on the other hand describe communication scenarios, where spatial *and* temporal correlation are concurrently of importance. The roles of the different parties are not limited to measurements, but consist of a read-out of information *as well* as a feed-forward of it. Quantum stochastic processes with nontrivial entanglement structure, in an operational setting, can be considered as a resource [78, 79], much in the same way as usual quantum channels [80, 81]. Consequently, having developed an understanding of the entanglement structure of spatio-temporal processes, a natural next step will be – based on the results we provided – to investigate how such genuine quantum correlations can be of use in complex quantum information tasks that require both common cause *and* direct cause causal relations for their successful implementation.

Finally, it is in principle possible to drop the requirement of causal order without a priori creating paradoxical situations. Such scenarios are described by process matrices, mathematical objects akin to quantum combs, but with the fundamental difference that they do not ensure global, but only local causality. While it has been shown that such more general processes can lead to advantages in quantum games [24, 82], the explicit origin of this advantage is hard to pin-point. Approaching this question based on the correlations that cannot persist in causally ordered combs but can occur in process matrices, might shed new light on the resourcefulness of these exotic objects.

**Note.** Upon publication of this paper we became aware of Ref. [83], where the authors proved the existence of separable combs that cannot be represented by means of a circuit containing entanglement breaking channels, thus completing our considerations of Sec. 5.

# Acknowledgements

We thank Marco Túlio Quintino for valuable discussions. We acknowledge funding from the DAAD Australia-Germany Joint Research Cooperation Scheme through the project 57445566. SM acknowledges funding from the Austrian Science Fund (FWF): ZK3 (Zukunftskolleg) and Y879-N27 (START project), the European Union's Horizon 2020 research and innovation programme under the Marie Skłodowska Curie grant agreement No 801110, and the Austrian Federal Ministry of Education, Science and Research (BMBWF). KM is supported through Australian Research Council Future Fellowship FT160100073 and Australian Research Council's Discovery Project DP210100597. ZX is supported through the Alexander von Humboldt Foundation. CS acknowledges support by the Austrian Science Fund (FWF): J 4258-N27 (Erwin-Schrödinger Programm) and Y879-N27 (START project), the ERC (Consolidator Grant 683107/TempoQ) and the Austrian Academy of Sciences. OG is supported by the ERC (Consolidator Grant 683107/TempoQ) and the Deutsche Forschungsgemeinschaft (DFG, German Research Foundation - Projektnummern 447948357 and 440958198).

# A Choi-Jamiołkowski isomorphism and the link product

In this Appendix, we review the basic properties of the CJI for maps and combs, as well as the link product. For any map $\mathcal{M} : \mathcal{B}(\mathcal{H}_X) \to \mathcal{B}(\mathcal{H}_Y)$, the CJI consists of letting $\mathcal{M}$ act on one half of an (unnormalized) maximally entangled state to map it onto a positive (Choi) matrix

$$M_{YX} := (\mathcal{M} \otimes \mathcal{I})[\widetilde{\Phi}_X^+] \in \mathcal{B}(\mathcal{H}_Y \otimes \mathcal{H}_X), \tag{53}$$

where $\widetilde{\Phi}^+ = \sum_{i,j=1}^{d_j} |ii\rangle\langle jj| \in \mathcal{B}(\mathcal{H}_X \otimes \mathcal{H}_{X'})$, and $\mathcal{H}_X \cong \mathcal{H}_{X'}$. Frequently encountered Choi matrices are the Choi matrix of the identity channel $\mathcal{I}_{X \to Y}[\rho] = \rho$, which is given by $\widetilde{\Phi}_{YX}^+$, and the Choi matrix of the partial trace $\text{tr}_X$, which is given by $\mathbb{1}_X$. It is straightforward to see that the map $\mathcal{M}$ is CPTP iff its Choi matrix $M_{YX}$ satisfies $M_{YX} \geq 0$ and $\text{tr}_Y(M_{YX}) = \mathbb{1}_X$.

In a similar vein the CJI exists for combs, with the difference that at each time $t_j$, one half of a (unnormalized) maximally entangled state is fed into the process (see Fig. 13). It is straightforward to check that the resulting quantum state satisfies the hierarchy of trace conditions of Eq. (4), while it has been shown in [19] that any positive matrix that satisfies these conditions can indeed be considered the Choi matrix of a quantum process. Finally, by simply inserting the respective definitions, one obtains

$$\mathcal{T}_{n+1}[\mathcal{M}_{x_1}, \dots, \mathcal{M}_{x_{n_1}}] = \text{tr}[\Upsilon_{n+1}(M_{x_1}^T \otimes \cdots \otimes M_{x_{n+1}}^T)], \tag{54}$$

the Born rule for temporal processes.

It is convenient, to not only phrase the respective maps in terms of their Choi matrices, but also to translate the concatenation of maps in terms of the respective Choi states. For example, the action of a map $\mathcal{M} : \mathcal{B}(\mathcal{H}_X) \to \mathcal{B}(\mathcal{H}_Y)$ on a state $\rho_X \in \mathcal{B}(\mathcal{H}_X)$ can be written in terms of Choi states [36] as

$$\mathcal{M}[\rho_X] = \text{tr}_X[M_{YX}(\mathbb{1}_Y \otimes \rho_X^T)] = M_{YX} \star \rho_X, \tag{55}$$

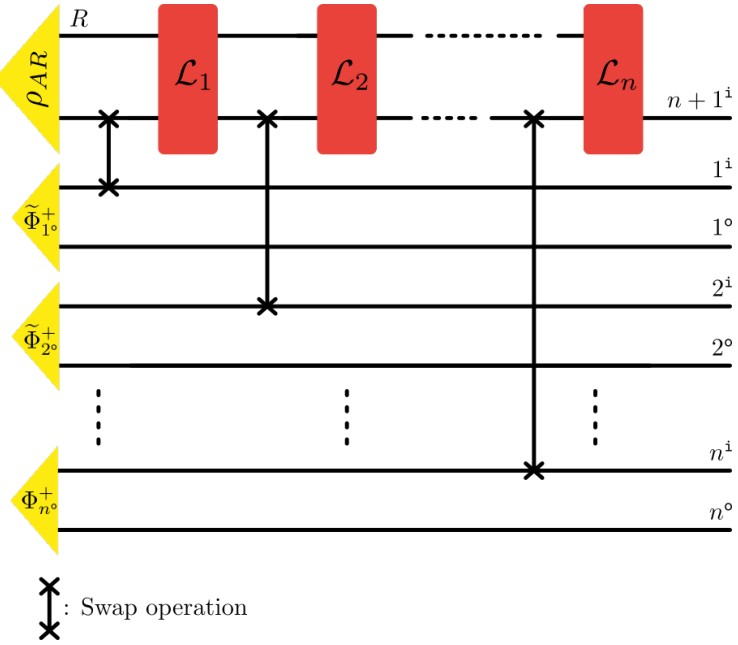

Figure 13: **CJI for combs.** At each time $t_j$, one half of a maximally entangled state it fed/swapped into the process. The resulting multipartite quantum state $\Upsilon'_{n+1} \in \mathcal{B}(\mathcal{H}_1^{\mathrm{i}} \otimes \mathcal{H}_1^{\mathrm{o}} \otimes \cdots \otimes \mathcal{H}_n^{\mathrm{o}} \otimes \mathcal{H}_{n+1}^{\mathrm{i}})$ contains all spatio-temporal correlations of the underlying process. For simpler notation, the CJI is generally phrased in terms of unnormalised maximally entangled states, i.e., $\Upsilon_{n+1} = d_1^{\mathrm{o}} \cdot d_2^{\mathrm{o}} \cdots d_n^{\mathrm{o}} \Upsilon'_{n_1}$

which, again, can be seen by direct insertion of the definition of $M_{XY}$. These considerations are generalised by the *link product* $\star$ [19], which translates the concatenation $\mathcal{G} \circ \mathcal{F}$ of two maps $\mathcal{F} : \mathcal{B}(\mathcal{H}_X) \to \mathcal{B}(\mathcal{H}_Y)$ and $\mathcal{G} : \mathcal{B}(\mathcal{H}_Y \otimes \mathcal{H}_Z)$ to an operation on their respective Choi states $F_{YX}$ and $G_{ZY}$. The Choi state of $\mathcal{G} \circ \mathcal{F}$ is then obtained via

$$F_{YX} \star G_{ZX} = \mathrm{tr}_Y[(F_{YX} \otimes \mathbb{1}_Z)(G_{ZY}^{\mathrm{T}_Y} \otimes \mathbb{1}_X)]. \tag{56}$$

For example, Eq. (55) can be understood as the Choi state of the concatenation $\mathcal{M} \circ \mathcal{R}$, where $\mathcal{R} : \mathbb{C} \to \mathcal{B}(\mathcal{H}_X)$ is the CPTP preparation map that prepares the state $\rho_X$ (i.e., the Choi matrix of $\mathcal{R}$ is equal to $\rho_X$). Put somewhat intuitively, the link product traces Choi matrices over the Hilbert spaces they are both defined on, and corresponds to a tensor product on the remaining spaces. From the definition, it can be directly seen that it satisfies

$$F \star G \star H = (F \star G) \star H = F \star (G \star H), \tag{57}$$

and it is commutative if each Hilbert space the respective matrices are defined on appears at most twice in the product [19] (which is always the case in this article). Additionally, the link product of positive matrices is positive as well. To simplify tracking of the involved spaces, it is helpful to add subscripts to the respective Choi matrices. For example, then, it is easy to see that a product of the form

$$F_{XYZ} \star G_W \star H_Y \star L_Z \tag{58}$$

is defined on $\mathcal{B}(\mathcal{H}_X \otimes \mathcal{H}_W)$ (as all the other spaces are traced over), and must be of the form $P_X \otimes G_W$, where $P_X = F_{XYZ} \star H_Y \star L_Z$. While every operation in this article could be written in terms of link products, it is sometimes more insightful to mix notation. For example, a term of the form $\mathrm{tr}(F_{XYZ} \star H_Y)$ could equivalently be written as $\mathbb{1}_{XZ} \star F_{XYZ} \star H_Y$ (since $\mathbb{1}_{XZ}$ is the Choi state of $\mathrm{tr}_{XZ}$).

# B  Proof of Proposition 6

In the following we provide the proof of Proposition 6:

**Proposition 6.** *If a comb $\Upsilon_{ABC}$ can be represented with an entanglement breaking channel on one of its wires, then it is separable in at least one possible splitting. In particular, an entanglement breaking channel on A or R implies $\mathcal{E}(A : BC) = 0$, on B implies $\mathcal{E}(B : AC) = 0$, and on C implies $\mathcal{E}(C : AB) = 0$.*

*Proof.* As the proofs for all cases proceed along the same lines, we explicitly show the Proposition 6 for EB channels on $C$ and $A$, with the other two cases following in a similar vein. First, for better book-keeping of the involved spaces, we will label the respective input and output spaces of the entanglement breaking maps differently, i.e, we have $\mathcal{N}^{\mathrm{EB}}_{C' \to C} : \mathcal{B}(\mathcal{H}_{C'}) \to \mathcal{B}(\mathcal{H}_C)$ and $\mathcal{N}^{\mathrm{EB}}_{A' \to A} : \mathcal{B}(\mathcal{H}_{A'}) \to \mathcal{B}(\mathcal{H}_A)$. Their Choi matrices are then denoted by $N^{\mathrm{EB}}_{C'C}$ and $N^{\mathrm{EB}}_{A'A}$, respectively. Additionally, in order for the resulting comb $\Upsilon_{ABC}$ to be defined on the spaces $ABC$, we add some primes to the spaces the building blocks of $\Upsilon_{ABC}$ are defined on (see Fig. 9). These relabellings are a mere notational aid and have no bearing on the results.

As each entanglement breaking channel can be represented as a measurement followed by a repreparation, we have $N^{\mathrm{EB}}_{XX'} = \sum_\alpha E_X^{(\alpha)\mathrm{T}} \otimes \xi_{X'}^{(\alpha)}$, where $\{E_X^{(\alpha)\mathrm{T}}\}_\alpha$ is a POVM, $\{\xi_{X'}^{(\alpha)}\}_\alpha$ is a collection of quantum states, and $X \in \{C, A\}$. With this, a comb $\Upsilon_{ABC}$ with an entanglement breaking channel on $C'$ (and building blocks $\rho_{AR}$, $L_{BRC'}$) is given by

$$\Upsilon_{ABC} = \rho_{AR} \star L_{BRC'} \star N^{\mathrm{EB}}_{C'C} = \sum_\alpha (\rho_{AR} \star L_{BRC'} \star E_{C'}^{(\alpha)}) \otimes \xi_C^{(\alpha)}, \tag{59}$$

which, as $\rho_{AR} \star L_{BRC'} \star E_{C'}^{(\alpha)} > 0$ is separable in the splitting $C : AB$. Analogously, a comb $\Upsilon_{ABC}$ resulting from a circuit with an entanglement breaking channel on the wire $A$ is given by

$$\Upsilon_{ABC} = \rho_{A'R} \star N^{\mathrm{EB}}_{A'A} \star L_{BRC} = \sum_\alpha \xi_A^{(\alpha)} \otimes (\rho_{A'R} \star E_{A'}^{(\alpha)} \star L_{BRC}), \tag{60}$$

which – for the same reason as above – is separable in the splitting $A : BC$. The remaining cases follow in a similar vein. $\qquad \square$

Note that Eqs. (59) and (60) provide the most general form of a comb that contains an EB channel on the respective wire.

# C  Proof of Lem. 7

Here, we provide a proof of Lem. 7 in the main text:

**Lemma 7.** *If a circuit can be represented with an EB channel on any two of the wires $\{A/R, B, C\}$ it is fully separable.*

*Proof.* We present an explicit proof for the case of entanglement breaking channels on the wires $A$ and $B$. the other cases follow analogously. In this case, the resulting comb $\Upsilon_{ABC}$ can be written as (see Fig. 9):

$$\Upsilon_{ABC} = \rho_{A'R} \star N^{\mathrm{EB}}_{AA'} \star L_{B'RC} \star M^{\mathrm{EB}}_{BB'}, \tag{61}$$

where $N^{\mathrm{EB}}_{A'A}$ and $M^{\mathrm{EB}}_{BB'}$ are entanglement breaking channels. Consequently, each of them can be decomposed as $N^{\mathrm{EB}}_{A'A} = \sum_\alpha E_{A'}^{(\alpha)\mathrm{T}} \otimes \eta_A^{(\alpha)}$ and $M^{\mathrm{EB}}_{BB'} = \sum_\beta F_B^{(\beta)} \otimes \tau_{B'}^{(\beta)}$, where $\{E_{A'}^{(\alpha)\mathrm{T}}\}_\alpha$ and $\{F_B^{(\beta)}\}_\beta$

are POVMs, and $\{\eta_A^{(\alpha)}\}_\alpha$ and $\{\tau_{B'}^{(\beta)}\}$ are sets of states. Inserting this into the above equation yields the fully separable comb

$$\Upsilon_{ABC} = \sum_{\alpha\beta} p(\alpha)\eta_A^{(\alpha)} \otimes F_B^{(\beta)} \otimes \xi_C^{(\alpha,\beta)}, \tag{62}$$

where $p(\alpha) = \mathrm{tr}(\rho_{A'R} E_{A'}^{(\alpha)\mathrm{T}})$, and $\xi_C^{(\alpha,\beta)} = (L_{B'RC} \star E_{A'}^{(\alpha)} \star \rho_{A'R} \star \tau_B^{(\beta)})/p(\alpha)$. □

## D   Circuit for a tripartite GME comb $\Upsilon_{ABC}$

Here, we provide an explicit construction for the quantum circuit that leads to the comb $\Upsilon_{ABC}$ provided in Eq (47), which is of the form

$$\Upsilon_{ABC} = |s_1\rangle\langle s_1| + |s_2\rangle\langle s_2|, \tag{63}$$

with

$$|s_1\rangle = \frac{|001\rangle}{\sqrt{2}} + \frac{|010\rangle + |100\rangle}{2} \quad \text{and} \quad |s_2\rangle = \frac{|110\rangle}{\sqrt{2}} + \frac{i(|101\rangle - |011\rangle)}{2}. \tag{64}$$

As mentioned in the main text, we have $\mathrm{tr}_C(\Upsilon_{ABC}) = \frac{1}{2}\mathbb{1}_A \otimes \mathbb{1}_B$. The circuit that leads $\Upsilon_{ABC}$ can be found directly by means of the Stinespring dilation; to this end, first, consider the purification

$$|\Upsilon_{ABCD}\rangle = |s_1\rangle |0_D\rangle + |s_2\rangle |1_D\rangle, \tag{65}$$

where $D$ is the auxiliary two-dimensional purification system. By construction, $|\Upsilon_{ABCD}\rangle$ is a purification of $\frac{1}{2}\mathbb{1}_A \otimes \mathbb{1}_B$, and so is $\Phi_{AA'}^+ \otimes \tilde{\Phi}_{BB'}^+$. As the latter purification is minimal, there is an isometry $V_{A'B'\to DC} := V$ on the purification spaces, such that

$$|\Upsilon_{ABCD}\rangle\langle\Upsilon_{ABCD}| = V(\Phi_{AA'}^+ \otimes \tilde{\Phi}_{BB'}^+)V^\dagger. \tag{66}$$

Consequently, we have

$$\Upsilon_{ABC} = \mathrm{tr}_D[V(\Phi_{AA'}^+ \otimes \tilde{\Phi}_{BB'}^+)V^\dagger]. \tag{67}$$

This equation corresponds to a quantum circuit of an open system dynamics, where one half of an unnormalized maximally entangled state is fed in. Concretely, said circuit starts in an initial state $\Phi_{AA'}^+$, the $A$ part of it swapped out of the process, while the $B'$ part of the state $\tilde{\Phi}_{BB'}^+$ is fed in. The $A'B'$ degrees of freedom undergo the unitary evolution $V$, and finally, the system $D$ is traced out (see Fig. 14 for a graphical representation). This corresponds to the Choi matrix of a comb with building blocks $\Phi_{AA'}^+$ (as the initial system-environment state) and $V$ (as the dynamics between $t_1$ and $t_2$). For the discussion of their properties, we relabel $\Phi_{AA'}^+ \to \Phi_{AR}^+$ and $B' \to B$ to abide by the notational convention set up in the main text. First, note that $\Phi_{AR}^+$ is entangled between the system $A$ and the environment $R$. The isometry $V$ can be constructed directly from Eqs (65) and (66). It is straightforward to see that

$$V|00\rangle = |10\rangle, \; V|01\rangle = \frac{1}{\sqrt{2}}(|00\rangle + i|11\rangle)$$

$$V|10\rangle = \frac{1}{\sqrt{2}}(|00\rangle - i|11\rangle), \; V|11\rangle = |01\rangle, \tag{68}$$

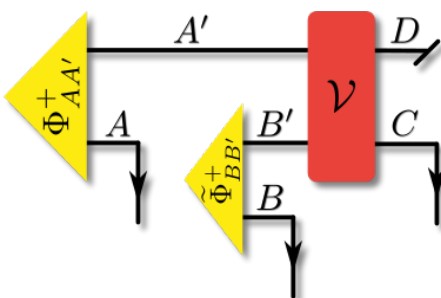

Figure 14: **Circuit that leads to $\Upsilon_{ABC}$.** See text for further details.

where the order of the spaces is $BR$ $(CD)$, e.g., the first equation reads $V|0_B 0_R\rangle = |1_C 0_D\rangle$. With this, we have

$$V = |10\rangle\langle 00| + \frac{1}{\sqrt{2}}(|00\rangle + i|11\rangle)\langle 01| + \frac{1}{\sqrt{2}}(|00\rangle - i|11\rangle)\langle 10| + |01\rangle\langle 11|, \quad (69)$$

which is not just an isometry, but a unitary matrix. The corresponding Choi matrix of the map $\mathcal{V}[\cdot] = V(\cdot)V^\dagger$ is proportional to a pure state, i.e., it is of the form $|v\rangle\langle v|$, with

$$|v\rangle = |1000\rangle + \frac{1}{\sqrt{2}}(|00\rangle + i|11\rangle)|01\rangle + \frac{1}{\sqrt{2}}(|00\rangle - i|11\rangle)|10\rangle + |0111\rangle), \quad (70)$$

where, the order of spaces is $CDBR$ (i.e., for example, we have the term $|1_C 0_D 0_B 0_R\rangle$). Finally, to be able to compare this map to the ones we discussed in Props. 1 – 3 the purification space $D$ needs to be traced over, yielding

$$
\begin{aligned}
L_{BRC} &:= \mathrm{tr}_D(|v\rangle\langle v|) \\
&= |001\rangle\langle 001| + \frac{1}{2}|010\rangle\langle 010| + \frac{1}{2}|011\rangle\langle 011| \\
&\quad + \frac{1}{2}|100\rangle\langle 100| + \frac{1}{2}|101\rangle\langle 101| + |110\rangle\langle 110| \\
&\quad + \frac{1}{\sqrt{2}}(|001\rangle\langle 010| + \mathrm{h.c.}) + \frac{1}{\sqrt{2}}(|001\rangle\langle 100| + \mathrm{h.c.}) \\
&\quad + \frac{i}{\sqrt{2}}(|011\rangle\langle 110| - \mathrm{h.c.}) + \frac{i}{\sqrt{2}}(|110\rangle\langle 101| - \mathrm{h.c.}) \\
&\quad + \frac{1}{2}(|010\rangle\langle 100| + \mathrm{h.c.}) - \frac{1}{2}(|011\rangle\langle 101| + \mathrm{h.c.})].
\end{aligned}
\quad (71)
$$

Here, and in what follows, we have switched the ordering of spaces to $BRC$ (i.e., for example, we have the term $|0_B 1_R 1_C\rangle\langle 1_B 0_R 1_C|$) to be consistent with the subscript of $L_{BRC}$.

As already mentioned, this tripartite entangled $\Upsilon_{ABC}$ can satisfy $\mathcal{E}(X:Y|z) > 0$ for all possible bipartitions. Here, we can directly check that its underlying building blocks indeed satisfy the necessary and sufficient conditions presented in the main text. Firstly, it is easy to check that, e.g., $|0\rangle\langle 0|_B \star L_{BRC}$ is proportional to an entangled state on $\mathcal{H}_R \otimes \mathcal{H}_C$, which, according to Prop. 1 is necessary for entanglement of the form $\mathcal{E}(A:C|b)$. Analogously, both $|0\rangle\langle 0|_R \star L_{BRC}$ and $|0\rangle\langle 0|_C \star L_{BRC}$ are proportional to entangled states, respectively. Consequently, the building blocks $\{\Phi_{AR}^+, L_{BRC}\}$ of $\Upsilon_{ABC}$ that we derived satisfy all the necessary conditions of Props. 1 – 3. While the Stinespring dilation of a comb – and as such its building blocks –is not unique [19], any pair $\{\rho_{AR}, L_{BRC}\}$ that satisfies $\rho_{AR} \star L_{BRC} = \Upsilon_{ABC}$ for the comb of Eq. (63) would satisfy the same conditions.

Additionally, we can show directly, that $\Upsilon_{ABC}$ satisfies the sufficient conditions for entanglement in the splittings $AB:C$ and $AC:B$ laid out in Sec. 4.1. There, we showed that if the

Choi matrix $L_{BRC} \star [\text{tr}_A(\rho_{AR})]$ is entangled, then $\Upsilon_{ABC}$ is entangled with respect to both of the splittings $AB : C$ and $AC : B$. In our case, $L_{BRC} \star \text{tr}_A(\rho_{AR})$ reads (with spaces arranged in the order $BC$)

$$\text{tr}_D(|v\rangle\langle v|) \star \text{tr}_A(\Phi_{AR}^+) = \frac{1}{2}\text{tr}_{DR}(|v\rangle\langle v|). \tag{72}$$

Then, from Eq. (71) we have

$$
\begin{aligned}
L_{BRC} \star \text{tr}_A(\rho_{AR}) = &\frac{1}{4}|00\rangle\langle 00| + \frac{3}{4}|10\rangle\langle 10| + \frac{3}{4}|01\rangle\langle 01| + \frac{1}{4}|11\rangle\langle 11| \\
&+ \frac{1+i}{2\sqrt{2}}|01\rangle\langle 10| + \frac{1-i}{2\sqrt{2}}|10\rangle\langle 01|,
\end{aligned}
\tag{73}
$$

which is (proportional to) an entangled state. Consequently, the maximally tripartite entangled comb $\Upsilon_{ABC}$ we found satisfies all of the necessary and sufficient conditions for bipartite entanglement we discussed in the main text.

## E    Non-zero tangle for the process in Eq. (50) with $n = 3$

In order to see that the process in Eq. (50) has indeed a non-zero tangle note first that if $\tau(\rho) = 0$, then also $\tau(A\rho A^\dagger / \text{tr}(A^\dagger A\rho)) = 0$ with $A = \otimes_i A_i$ and $A_i \in GL(2, \mathbb{C})$ [62, 65]. However, as we will show there exist local invertible operators that transform $\Upsilon^{(3)}$ to a state which has non-zero tangle and therefore also $\tau(\Upsilon^{(3)}) \neq 0$. These operators can be chosen to be $A_1 = A_2 = \text{Diag}(1/\sqrt{\alpha}, \sqrt{\alpha})$ and $A_3 = \text{Diag}(\alpha, 1/\alpha)$ with $0 < \alpha < 1/\sqrt{3}$ and $\text{Diag}(x, y)$ denoting a diagonal matrix with entries $x$ and $y$. The transformed state is of the form

$$\frac{A\Upsilon^{(3)}A^\dagger}{\text{tr}(A\Upsilon^{(3)}A^\dagger)} = \frac{1}{1+\alpha^2}|GHZ_3\rangle\langle GHZ_3| + \frac{\alpha^2}{2(1+\alpha^2)}(|01\rangle\langle 01| + |10\rangle\langle 10|) \otimes |0\rangle\langle 0|. \tag{74}$$

Note that for convenience we used here the normalization for states. This state can be detected for example for $\alpha < 1/\sqrt{3}$ to be not in the W-class by using the witness [63] $W = 3/4\mathbb{1} - |GHZ_3\rangle\langle GHZ_3|$ and therefore has non-zero tangle. Hence, also $\Upsilon^{(3)}$ is not in the W-class.

## F    Circuit for a GME comb on three steps

Here, we provide an explicit example for a simple circuit that yields a genuinely multipartite entangled comb on three times, i.e., on four Hilbert spaces. In the case we discuss, the system and the environment are a qubit, respectively, and the four-legged resulting comb has an initial input leg ($A$) at time $t_1$, a middle slot (BC) at $t_2$, where general CP maps can be performed, and a final output leg ($D$) at time $t_3$ (see Fig. 15). As an intermediate system environment in between times we choose the square root of the swap operator $\sqrt{\mathcal{S}}[\rho] = \sqrt{S}\rho\sqrt{S}^\dagger$, where $S$ is the Swap gate and $\sqrt{S}$ is represented by

$$\sqrt{S} = \frac{1}{2}\begin{pmatrix} 2 & 0 & 0 & 0 \\ 0 & 1+i & 1-i & 0 \\ 0 & 1-i & 1+i & 0 \\ 0 & 0 & 0 & 2 \end{pmatrix} \tag{75}$$

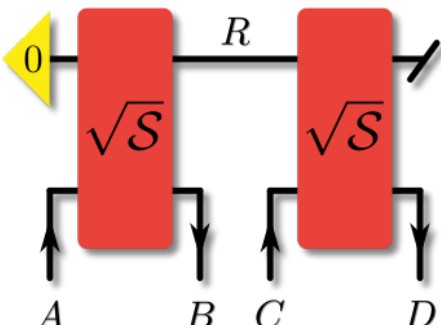

Figure 15: **4-partite genuinely multipartite entangled comb.** See text for details.

in the standard system-environment basis $\{|00\rangle, |01\rangle, |10\rangle, |11\rangle\}$. Choosing $|0\rangle$ for the initial state of the environment, and tracing out the environmental degrees of freedom at $t_3$ then yields the comb

$$\Upsilon_{ABCD} = \frac{1}{4}(5\,|s_1\rangle\langle s_1| + 3\,|s_2\rangle\langle s_2|), \tag{76}$$

where

$$|s_1\rangle \propto -2i\,|0000\rangle + (1-i)\,|0011\rangle - |1001\rangle + (1-i)\,|1100\rangle + |1111\rangle, \tag{77}$$
$$|s_2\rangle \propto (1-i)\,|0010\rangle + |1000\rangle - |1001\rangle + (1-i)\,|1011\rangle + |1110\rangle, \tag{78}$$

and the ordering of spaces is $ABCD$, i.e, for example, $|1_A 0_B 0_C 1_D\rangle$. It is straightforward to see that the comb $\Upsilon_{ABCD}$ of Eq. (76) is positive and satisfies the causality constraints of Eq. (4). Finally, using the SDP (12), it is easy to check that $\Upsilon_{ABCD}$ is indeed genuinely multipartite entangled.

Importantly, choosing the swap operator $\mathcal{S}$ instead of $\sqrt{\mathcal{S}}$ would not have lead to a GME comb. Rather, the Swap would lead to identity channels between the respective inputs and outputs, leading to an overall non-GME comb of the form

$$\Upsilon_{ABCD}^{(\mathcal{S})} = 2\Phi_{AD}^+ \otimes |0\rangle\langle 0|_B \otimes \mathbb{1}_C, \tag{79}$$

where $2\Phi_{AD}^+$ is the Choi state of the identity channel between $A$ and $D$.

# G  GME processes with vanishing conditional entanglement

Following the steps laid out in Sec. 8, we find the following GME comb with vanishing conditional entanglement:

$\Upsilon_{ABC}$

$$\approx 10^{-2} \begin{pmatrix} 13.8 & 2.1+2i & 1.2+1i & -11.5-4.8i & 4+1.5i & 10.9-5.6i & -0.1+0.3i & 2.1-2.7i \\ 2.1-2i & 36.2 & -15.7+1.2i & -1.2-0.1i & -15.4+3.5i & -0.4-1.5i & -4.5-2.6i & 0.1-0.3i \\ 1.2-0.1i & -15.7-1.2i & 24.9 & 1.2+0.3i & -3.2-11.9i & 1.7+0.1i & -0.4-0.3i & 12.1+1.5i \\ -11.5+4.8i & -1.2+0.1i & 1.2-0.3i & 25.1 & -0.3-1.6i & 3.2+11.9i & 14.9-5.2i & 0.4+0.3i \\ 0.4-1.5i & -15.4-3.5i & -3.2+11.9i & -0.3+1.6i & 25 & 0.3+1.4i & -0.3-0.5i & -11.9+2.6i \\ 10.9+5.6i & -0.4+1.5i & 1.7-0.1i & 3.2-11.9i & 0.3-1.4i & 25 & 15.7+0.7i & 0.3+0.5i \\ -0.1-0.3i & -4.5+2.6i & -0.4+0.3i & 14.9+5.2i & -0.3+0.5i & 15.7-0.7i & 35.8 & 3.2+3.7i \\ 02.1+2.7i & 0.1+0.3i & 12.1-1.5i & 0.4-0.3i & -11.9-2.6i & 0.3-0.5i & 3.2-3.7i & 14.2 \end{pmatrix}, \tag{80}$$

where $\Upsilon_{ABC}$ is represented in the standard product basis, i.e., $|\bar{0}\rangle = |0_A 0_B 0_C\rangle, |\bar{1}\rangle = |0_A 0_B 1_C\rangle, |\bar{2}\rangle = |0_A 1_B 0_C\rangle, \ldots$. Up to numerical precision, the above $\Upsilon_{ABC}$ is a proper comb, as it is positive semidefinite, satisfies $\mathrm{tr}(\Upsilon_{ABC}) = d_B = 2$, and we have $\mathrm{tr}_C(\Upsilon_{ABC}) = \frac{1}{2}\mathbb{1}_{AB}$.

$\Upsilon_{ABC}$ is a good candidate for a GME comb that has vanishing conditional entanglement. As all the involved subsystems are qubits, the fact that this is indeed the case can be shown by applying the PPT criterion to the (normalised) post-measurement states. Any pure qubit state $|\Psi\rangle$ can be parameterized in terms of Pauli matrices as

$$|\Psi\rangle\langle\Psi| = \tfrac{1}{2}\big[\mathbb{1} + \cos(\vartheta)\cos(\varphi)\sigma_x + \cos(\vartheta)\sin(\varphi)\sigma_y + \sin(\vartheta)\sigma_z\big]. \tag{81}$$

With this, the respective conditioned combs $\text{tr}_X(\Upsilon_{ABC} |\Psi\rangle\langle\Psi|_X)$ for arbitrary (pure) projective 'measurements' on $X \in \{A, B, C\}$ can be computed[4].

As the resulting conditioned combs are defined on two qubits, their entanglement can be decided by means of the PPT criterion; while it cannot be straightforwardly shown analytically that the resulting conditioned combs are indeed PPT, we can check the positivity of their partial transpose numerically, for a sufficiently large number of angles. Here, we randomly choose $5 \times 10^5$ uniformly distributed pairs $(\varphi, \vartheta) \in [0, 2\pi] \times [0, \pi]$ and compute the minimal eigenvalue of the (normalised) partial transpose $\rho_{YZ}^{T_Y}(\vartheta, \varphi)$ of the respective reduced states conditioned states. The minimal obtained eigenvalues we found are given by

$$\lambda_{\min}^{AB} = 0.0124, \lambda_{\min}^{BC} = 0.0187, \text{ and } \lambda_{\min}^{AC} = 0.0193. \tag{82}$$

Given that each of these values is well above zero, the resulting conditional states are all separable, and since the angles we sampled cover the relevant parameter space sufficiently finely, we conclude that the conditional states are indeed separable for *all* projective measurements. Since any POVM element (or – in the case of $B$ – any state) can be written up to normalization as a convex combination of pure states, this then implies that the GME comb of Eq. (80) displays vanishing conditional entanglement for *all* conceivable measurements (or preparations, in the case of $B$).

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
