# Peer review of "Genuine Multipartite Entanglement in Time"

_SciPost Physics, doi:SciPost Phys. 10, 141 (2021)_

## Round 2 · Referee Report · Anonymous (Referee 1) · 2021-5-9

Report

The submitted manuscript entitled "Genuine Multipartite Entanglement in Time" by S. Milz et al. addresses the question of obtaining genuine multipartite entanglement in temporal processes. In this work using the so-called quantum comb procedure temporal processes are transformed to the spatial domain with the restriction that temporal correlations have to admit causal ordering (unlike spatial correlations). This mapping allows the Authors to apply established tools in the theory of multipartite entanglement to reveal the structure of temporal quantum correlations.

In my view this is a very nice piece of work which unifies several key concepts about multiple-time temporal processes. In particular, the paper gives conditions for the presence of "bipartite entanglement" in different splittings associated with two-time processes. The studied scenarios are linked to the well-developed concepts of quantum memory, entanglement breaking channels and EPR steering. Furthermore, explicit examples are provided in the two-time scenario for the enigmatic GHZ and W-states.

In my view, the paper is very well written and the results obtained are correct. Also, all the presented examples are elegant. I highly recommend the paper for publication.

Minor technical comment/question: - Page 32: second line above the bottom of the page: "non-commuting" -> "non-communicating" - Section 6: The Authors show the existence of processes that are entangled in any bipartition, yet these processes are not genuinely tripartite entangled. In other words, it is shown that entanglement in every splitting E(A:BC)>0, E(C:AB)>0, E(B:AC)>0 does not imply in general genuinely tripartite entanglement. I would like to ask whether the stronger conditions for entanglement in the three marginals E(A:B)>0, E(B:C)>0, E(A:C)>0 would still imply genuinely tripartite entanglement for temporal processes.

  • validity: top
  • significance: top
  • originality: top
  • clarity: top
  • formatting: perfect
  • grammar: perfect

Author:  Simon Milz  on 2021-05-18  [id 1431]

(in reply to Report 1 on 2021-05-09)
Category:
answer to question

We thank the Referee for their careful reading of our manuscript and their positive evaluation. We have changed the typo on p. 32. With respect to the question of whether the stronger condition E(A:B)>0, E(B:C)>0, and E(A:C)>0 implies genuine multipartite entanglement for combs: Due to the causality constraints on combs, E(A:B) always vanishes. On the other hand, Eq. (45) in the manuscript already provides a process that satisfies E(A:B)>0 and E(B:C)>0 but is biseparable. We have added a sentence below Eq. (45) to emphasize that the provided process not only constitutes an example for a comb that satisfies E(A:BC)>0, E(B:AC)>0, and E(C:AB)>0 but is not GME, but at the same time also shows that the conceptually different question of whether E(A:B)>0 and E(B:C)>0 imply GME for combs has a negative answer.

Anonymous on 2021-05-21  [id 1445]

(in reply to Simon Milz on 2021-05-18 [id 1431])

I highly recommend the paper for publication in SciPost Physics.

---

## Round 2 · Referee Report · Anonymous (Referee 2) · 2021-5-10

Report

The authors present a comprehensive view of genuine multipartite entanglement in time using the quantum comb formalism. They mainly concentrate on processes that are probed at two time-points. It allows to effectively describe the system as three qubits. In the next sections, it is shown how to construct a necessary and sufficient condition for bipartite entanglement in different partitions. The paper is accompanied by examples of processes corresponding to the well-known GHZ and W type states.

The work is excellently written and, as I wrote earlier, is a comprehensive study of the problem (with all the details). The structure of the work resembles a PhD dissertation. The paper will be of interest to the quantum information community and the way it is presented will make it easy to understand at any level of experience.

I have no comments and recommend publishing the manuscript as it is.
  • validity: top
  • significance: high
  • originality: top
  • clarity: top
  • formatting: excellent
  • grammar: perfect

Author:  Simon Milz  on 2021-05-18  [id 1433]

(in reply to Report 2 on 2021-05-10)
Category:
remark

We thank the Referees for their careful reading as well as their positive evaluation of our manuscript.

---

## Round 3 · Referee Report · Anonymous · 2021-5-25

Report

I highly recommend the paper for publication in SciPost Physics.

---

## Round 3 · List of Changes

We followed the referees' suggestions and fixed a typo in the conclusions (commuting -> communicating) and emphasized in Sec. 6 that entanglement in the reduced combs does not imply genuine multipartite entanglement in the overall comb

---

## Editorial Decision

published